# EQUILIBRIUM PROPAGATION WITH CONTINUAL WEIGHT UPDATES

## ABSTRACT

Equilibrium Propagation (EP) is a learning algorithm that bridges Machine Learning and Neuroscience, by computing gradients closely matching those of Backpropagation Through Time (BPTT), but with a learning rule local in space. Given an input $x$ and associated target $y$, EP proceeds in two phases: in the first phase neurons evolve freely towards a first steady state; in the second phase output neurons are nudged towards $y$ until they reach a second steady state. However, in existing implementations of EP, the learning rule is not local in time: the weight update is performed after the dynamics of the second phase have converged and requires information of the first phase that is no longer available physically. In this work, we propose a version of EP named Continual Equilibrium Propagation (C-EP) where neuron and synapse dynamics occur simultaneously throughout the second phase, so that the weight update becomes local in time. Such a learning rule local both in space and time opens the possibility of an extremely energy efficient hardware implementation of EP. We prove theoretically that, provided the learning rates are sufficiently small, at each time step of the second phase the dynamics of neurons and synapses follow the gradients of the loss given by BPTT (Theorem 1). We demonstrate training with C-EP on MNIST and generalize C-EP to neural networks where neurons are connected by asymmetric connections. We show through experiments that the more the network updates follows the gradients of BPTT, the best it performs in terms of training. These results bring EP a step closer to biology by better complying with hardware constraints while maintaining its intimate link with backpropagation.

## 1 INTRODUCTION

A motivation for deep learning is that a few simple principles may explain animal intelligence and allow us to build intelligent machines, and learning paradigms must be at the heart of such principles, creating a synergy between neuroscience and Artificial Intelligence (AI) research. In the deep learning approach to AI (LeCun et al., 2015), backpropagation thrives as the most powerful algorithm for training artificial neural networks. Unfortunately, its implementation on conventional computer or dedicated hardware consumes more energy than the brain by several orders of magnitude (Strubell et al., 2019). One path towards reducing the gap between brains and machines in terms of power consumption is by investigating alternative learning paradigms relying on locally available information, which would allow radically different hardware implementations: such local learning rules could be used for the development of extremely energy efficient learning-capable hardware. Investigating such bioplausible learning schemes with real-world applicability is therefore of interest not only for neuroscience, but also for developing neuromorphic computing hardware that takes inspiration from information-encoding, dynamics and topology of the brain to reach fast and energy efficient AI (Ambrogio et al., 2018; Romera et al., 2018). In these regards, Equilibrium Propagation (EP) is an alternative style of computation for estimating error gradients that presents significant advantages (Scellier and Bengio, 2017).

EP belongs to the family of contrastive Hebbian learning (CHL) algorithms (Ackley et al., 1985; Movellan, 1991; Hinton, 2002) and therefore benefits from an important feature of these algorithms: neural dynamics and synaptic updates depend solely on information that is locally available. As a CHL algorithm, EP applies to convergent RNNs, i.e. RNNs that are fed by a static input and converge to a steady state. Training such a convergent RNN consists in adjusting the weights so that the

steady state corresponding to an input $x$ produces output values close to associated targets $y$. CHL algorithms proceed in two phases: in the first phase, neurons evolve freely without external influence and settle to a (first) steady state; in the second phase, the values of output neurons are influenced by the target $y$ and the neurons settle to a second steady state. CHL weight updates consist in a Hebbian rule strengthening the connections between co-activated neurons at the first steady state, and an anti-Hebbian rule with opposite effect at the second steady state. A difference between Equilibrium Propagation and standard CHL algorithms is that output neurons are not clamped in the second phase but elastically pulled towards the target $y$.

A second key property of EP is that, unlike CHL and other related algorithms, it is intimately linked to backpropagation. It has been shown that synaptic updates in EP follow gradients of recurrent backpropagation (RBP) (Scellier and Bengio, 2019) and backpropagation through time (BPTT) (Ernoult et al., 2019). This makes it especially attractive to bridge the gap between neural networks developed by neuroscientists, neuromorphic researchers and deep learning researchers.

Nevertheless, the bioplausibility of EP still undergoes two major limitations. First, although EP is local in space, it is non-local in time. In all existing implementations of EP the weight update is performed after the dynamics of the second phase have converged, when the first steady state is no longer physically available. Thus the first steady state has to be artificially stored. Second, the network dynamics have to derive from a primitive function, which is equivalent to the requirement of symmetric weights in the Hopfield model. These two requirements are biologically unrealistic and also hinder the development of efficient EP computing hardware.

In this work, we propose an alternative implementation of EP (called C-EP) which features temporal locality, by enabling synaptic dynamics to occur throughout the second phase, simultaneously with neural dynamics. We then address the second issue by adapting C-EP to systems having asymmetric synaptic connections, taking inspiration from Scellier et al. (2018) ; we call this modified version C-VF.

More specifically, the contributions of the current paper are the following:

- We introduce *Continual Equilibrium Propagation* (C-EP, Section 3.1-3.2), a new version of EP with continual weight updates: the weights of the network are adjusted continually in the second phase of training using local information in space and time. Neuron steady states do not need to be stored after the first phase, in contrast with standard EP where a global weight update is performed at the end of the second phase. Like standard EP, the C-EP algorithm applies to networks whose synaptic connections between neurons are assumed to be symmetric and tied.

- We show mathematically that, provided that the changes in synaptic strengths are sufficiently slow (i.e. the learning rates are sufficiently small), at each time step of the second phase the dynamics of neurons and synapses follow the gradients of the loss obtained with BPTT (Theorem 1 and Fig. 2, Section 3.3). We call this property the *Gradient Descending Dynamics* (GDD) property, for consistency with the terminology used in Ernoult et al. (2019).

- We demonstrate training with C-EP on MNIST, with accuracy approaching the one obtained with standard EP (Section 4.2).

- Finally, we adapt our C-EP algorithm to the more bio-realistic situation of a neural network with asymmetric connections between neurons. We call this modified version C-VF as it is inspired by the *Vector Field* method proposed in Scellier et al. (2018). We demonstrate this approach on MNIST, and show numerically that the training performance is correlated with the satisfaction of Gradient Descending Dynamics (Section 4.3).

For completeness, we also show how the Recurrent Backpropagation (RBP) algorithm of Almeida (1987); Pineda (1987) relates to C-EP, EP and BPTT. We illustrate the equivalence of these four algorithms on a simple analytical model (Fig. 3) and we develop their relationship in Appendix A.

## 2 BACKGROUND: CONVERGENT RNNS AND EQUILIBRIUM PROPAGATION

**Convergent RNNs With Static Input.** We consider the supervised setting, where we want to predict a target $y$ given an input $x$. The model is a recurrent neural network (RNN) parametrized by $\theta$

and evolving according to the dynamics:

$$s_{t+1} = F(x, s_t, \theta).$$ (1)

$F$ is the transition function of the system. Assuming convergence of the dynamics before time step $T$, we have $s_T = s_*$ where $s_*$ is the steady state of the network characterized by

$$s_* = F(x, s_*, \theta).$$ (2)

The number of timesteps $T$ is a hyperparameter that we choose large enough so that $s_T = s_*$ for the current value of $\theta$. The goal is to optimize the parameter $\theta$ in order to minimize a loss:

$$\mathcal{L}^* = \ell(s_*, y).$$ (3)

Algorithms that optimize the loss $\mathcal{L}^*$ for RNNs include Backpropagation Through Time (BPTT) and the Recurrent Backpropagation (RBP) algorithm of Almeida (1987); Pineda (1987), presented in Appendix B.

**Equilibrium Propagation (EP).** EP (Scellier and Bengio, 2017) is a learning algorithm that computes the gradient of $\mathcal{L}^*$ in the particular case where the transition function F derives from a scalar function $\Phi$, i.e. with $F$ of the form $F(x, s, \theta) = \frac{\partial \Phi}{\partial s}(x, s, \theta)$. The algorithm consists in two phases (see Alg. 1 of Fig. 1). During the first phase, the network follows a sequence of states $s_1, s_2, s_3 \ldots$ and converges to a steady state denoted $s_*$. In the second phase, an extra term $\beta \frac{\partial \ell}{\partial s}$ pertubs the dynamics of the neurons (where $\beta > 0$ is a scalar hyperparameter): starting from the steady state $s_0^\beta = s_*$, the network follows a second sequence of states $s_1^\beta, s_2^\beta, s_3^\beta \ldots$ and converges to a new steady state denoted $s_*^\beta$. Scellier and Bengio (2017) have shown that the gradient of the loss $\mathcal{L}^*$ can be estimated based on the two steady states $s_*$ and $s_*^\beta$. Specifically, in the limit $\beta \to 0$,

$$\frac{1}{\beta} \left( \frac{\partial \Phi}{\partial \theta}(x, s_*^\beta, \theta) - \frac{\partial \Phi}{\partial \theta}(x, s_*, \theta) \right) \to -\frac{\partial \mathcal{L}^*}{\partial \theta}.$$ (4)

## 3 Equilibrium Propagation with Continual Weight Updates (C-EP)

This section presents the main theoretical contributions of this paper. We introduce a new algorithm to optimize $\mathcal{L}^*$ (Eq. 3): a new version of EP with continual parameter updates that we call C-EP. Unlike typical machine learning algorithms (such as BPTT, RBP and EP) in which the weight updates occur after all the other computations in the system are performed, our algorithm offers a mechanism in which the weights are updated continuously as the states of the neurons change.

### 3.1 From EP to C-EP: An intuition behind continual weight updates

The key idea to understand how to go from EP to C-EP is that the gradient of EP appearing in Eq. (4) reads as the following telescopic sum:

$$\underbrace{\frac{1}{\beta} \left( \frac{\partial \Phi}{\partial \theta}(x, s_*^\beta, \theta) - \frac{\partial \Phi}{\partial \theta}(x, s_*, \theta) \right)}_{\text{global parameter gradient in EP}} = \sum_{t=1}^{\infty} \underbrace{\frac{1}{\beta} \left( \frac{\partial \Phi}{\partial \theta}(x, s_t^\beta, \theta) - \frac{\partial \Phi}{\partial \theta}(x, s_{t-1}^\beta, \theta) \right)}_{\text{parameter gradient at time t in C-EP}}.$$ (5)

In Eq. (5) we have used that $s_0^\beta = s_*$ and $s_t^\beta \to s_*^\beta$ as $t \to \infty$. Here lies the very intuition of continual updates motivating this work; instead of keeping the weights fixed throughout the second phase and updating them at the end of the second phase based on the steady states $s_*$ and $s_*^\beta$, as in EP (Alg. 1 of Fig. 1), the idea of the C-EP algorithm is to update the weights at each time $t$ of the second phase between two consecutive states $s_{t-1}^\beta$ and $s_t^\beta$ (Alg. 2 of Fig. 1). One key difference in C-EP compared to EP though, is that, in the second phase, the weight update at time step $t$ influences the neural states at time step $t + 1$ in a nontrivial way, as illustrated in the computational graph of Fig. 2. In the next subsection we define C-EP using notations that explicitly show this dependency.

**Algorithm 1** EP

*Input*: $x, y, \theta, \beta, \eta$.
*Output*: $\theta$.

1: $s_0 \leftarrow 0$          ▷ First Phase
2: **repeat**
3:     $s_{t+1} \leftarrow \frac{\partial \Phi}{\partial s}(x, s_t, \theta)$
4: **until** $s_t = s_*$
5: Store $s_*$
6: $s_0^\beta \leftarrow s_*$          ▷ Second Phase
7: **repeat**
8:     $s_{t+1}^\beta \leftarrow \frac{\partial \Phi}{\partial s}\left(x, s_t^\beta, \theta\right) - \beta \frac{\partial \ell}{\partial s}\left(s_t^\beta, y\right)$
9: **until** $s_t^\beta = s_*^\beta$
10:         ▷ Global Parameter Update
11: $\theta \leftarrow \theta + \frac{\eta}{\beta}\left(\frac{\partial \Phi}{\partial \theta}\left(s_*^\beta, \theta\right) - \frac{\partial \Phi}{\partial \theta}\left(s_*, \theta\right)\right)$

**Algorithm 2** C-EP (with simplified notations)

*Input*: $x, y, \theta, \beta, \eta$.
*Output*: $\theta$.

1: $s_0 \leftarrow 0$          ▷ First Phase
2: **repeat**
3:     $s_{t+1} \leftarrow \frac{\partial \Phi}{\partial s}(x, s_t, \theta)$
4: **until** $s_t = s_*$
5: $s_0^\beta \leftarrow s_*$          ▷ Second Phase
6: **repeat**
7:     $s_{t+1}^\beta \leftarrow \frac{\partial \Phi}{\partial s}\left(x, s_t^\beta, \theta\right) - \beta \frac{\partial \ell}{\partial s}\left(s_t^\beta, y\right)$
8:         ▷ Parameter Update at Time $t$
9:     $\theta \leftarrow \theta + \frac{\eta}{\beta}\left(\frac{\partial \Phi}{\partial \theta}\left(s_{t+1}^\beta\right) - \frac{\partial \Phi}{\partial \theta}\left(s_t^\beta\right)\right)$
10: **until** $s_t^\beta$ and $\theta$ are converged.

Figure 1: **Left.** Pseudo-code of EP. This is the version of EP for discrete-time dynamics introduced in Ernoult et al. (2019). **Right.** Pseudo-code of C-EP with simplified notations (see section 3.2 for a formal definition of C-EP). **Difference between EP and C-EP.** In EP, one global parameter update is performed at the end of the second phase ; in C-EP, parameter updates are performed throughout the second phase. Eq. 5 shows that the continual updates of C-EP add up to the global update of EP.

## 3.2 DESCRIPTION OF THE C-EP ALGORITHM

The first phase of C-EP is the same as that of EP (see Fig. 1). In the second phase of C-EP the parameter variable is regarded as another dynamic variable $\theta_t$ that evolves with time $t$ along with $s_t$. The dynamics of $s_t$ and $\theta_t$ in the second phase of C-EP depend on the values of the two hyperparameters $\beta$ (the hyperparameter of influence) and $\eta$ (the learning rate), therefore we write $s_t^{\beta,\eta}$ and $\theta_t^{\beta,\eta}$ to show explicitly this dependence. With now both the neurons and the synapses evolving in the second phase, the dynamic variables $s_t^{\beta,\eta}$ and $\theta_t^{\beta,\eta}$ start from $s_0^{\beta,\eta} = s_*$ and $\theta_0^{\beta,\eta} = \theta$ and follow:

$$\forall t \geq 0: \quad \begin{cases} s_{t+1}^{\beta,\eta} = \dfrac{\partial \Phi}{\partial s}\left(x, s_t^{\beta,\eta}, \theta_t^{\beta,\eta}\right) - \beta \dfrac{\partial \ell}{\partial s}\left(s_t^{\beta,\eta}, y\right), \\ \theta_{t+1}^{\beta,\eta} = \theta_t^{\beta,\eta} + \dfrac{\eta}{\beta}\left(\dfrac{\partial \Phi}{\partial \theta}\left(x, s_{t+1}^{\beta,\eta}, \theta_t^{\beta,\eta}\right) - \dfrac{\partial \Phi}{\partial \theta}\left(x, s_t^{\beta,\eta}, \theta_t^{\beta,\eta}\right)\right). \end{cases} \quad (6)$$

The difference in C-EP compared to EP is that the value of the parameter used to update $s_{t+1}^{\beta,\eta}$ in Eq. (6) is the current $\theta_t^{\beta,\eta}$, not $\theta$. Provided the learning rate $\eta$ is small enough, i.e. the synapses are slow compared to the neurons, this effect is weak. Intuitively, in the limit $\eta \to 0$, the parameter changes are negligible so that $\theta_t^{\beta,\eta}$ can be approximated by its initial value $\theta_0^{\beta,\eta} = \theta$. Under this approximation, the dynamics of $s_t^{\beta,\eta}$ in C-EP and the dynamics of $s_t^\beta$ in EP are the same. See Fig. 3 for a simple example, and Appendix A.3 for a proof in the general case.

## 3.3 GRADIENT-DESCENDING DYNAMICS (GDD)

Now we prove that, provided the hyperparameter $\beta$ and the learning rate $\eta$ are small enough, the dynamics of the neurons and the weights given by Eq. (6) follow the gradients of BPTT (Theorem 1 and Fig. 2). For a formal statement of this property, we *define* the *normalized (continual) updates* of C-EP, as well as the gradients of the loss $\mathcal{L} = \ell(s_T, y)$ after $T$ time steps, computed with BPTT:

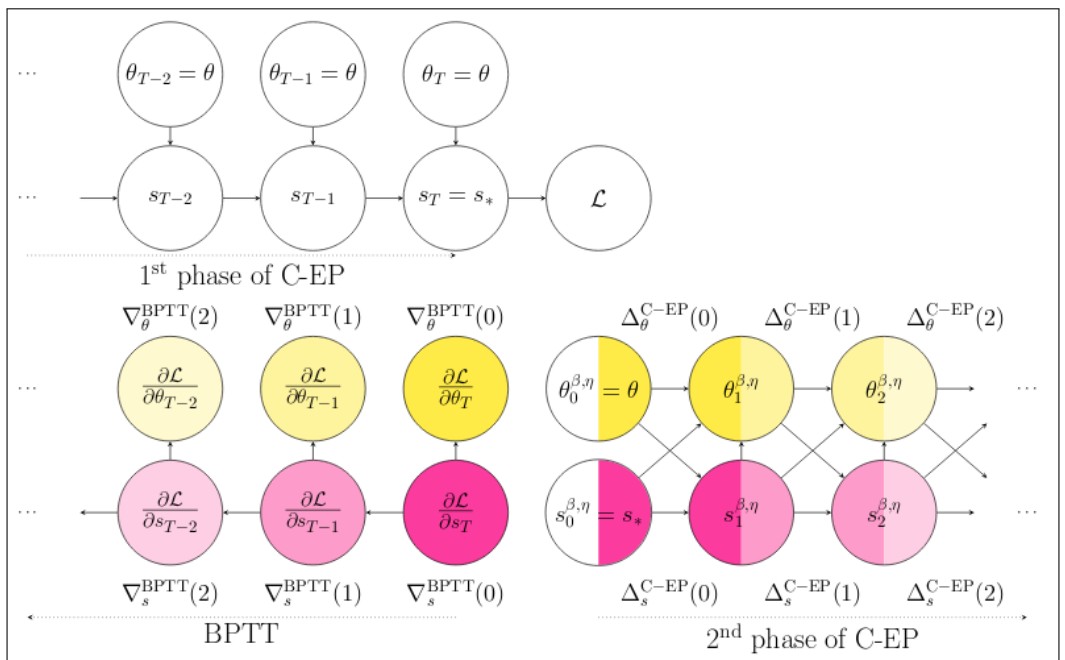

Figure 2: **Gradient-Descending Dynamics (GDD, Theorem 1).** In the second phase of Continual Equilibrium Prop (C-EP), the dynamics of neurons and synapses descend the gradients of BPTT, i.e. $\Delta^{\mathrm{C-EP}}(t) = -\nabla^{\mathrm{BPTT}}(t)$. The colors illustrate when corresponding computations are realized in C-EP and BPTT. **Top left.** 1$^{\mathrm{st}}$ phase of C-EP with static input $x$ and target $y$. The final state $s_T$ is the steady state $s_*$. **Bottom left.** Backprop through time (BPTT). **Bottom right.** 2$^{\mathrm{nd}}$ phase of C-EP. The starting state $s_0^{\beta,\eta}$ is the final state of the forward-time pass, i.e. the steady state $s_*$.

$$\left\{ \begin{array}{l} \Delta_s^{\mathrm{C-EP}}(\beta,\eta,t) = \dfrac{1}{\beta}\left(s_{t+1}^{\beta,\eta} - s_t^{\beta,\eta}\right), \\[2mm] \Delta_\theta^{\mathrm{C-EP}}(\beta,\eta,t) = \dfrac{1}{\eta}\left(\theta_{t+1}^{\beta,\eta} - \theta_t^{\beta,\eta}\right), \end{array} \right. \qquad \left\{ \begin{array}{l} \nabla_s^{\mathrm{BPTT}}(t) = \dfrac{\partial \mathcal{L}}{\partial s_{T-t}}, \\[2mm] \nabla_\theta^{\mathrm{BPTT}}(t) = \dfrac{\partial \mathcal{L}}{\partial \theta_{T-t}}. \end{array} \right. \qquad (7)$$

More details about $\mathcal{L}$ and the gradients $\nabla_s^{\mathrm{BPTT}}(t)$ and $\nabla_\theta^{\mathrm{BPTT}}(t)$ are provided in Appendix B. Note that injecting Eq. (6) in Eq. (7), the normalized updates of C-EP consequently read:

$$\Delta_\theta^{\mathrm{C-EP}}(\beta,\eta,t) = \frac{1}{\beta}\left( \frac{\partial \Phi}{\partial \theta}\left(x, s_{t+1}^{\beta,\eta}, \theta_t^{\beta,\eta}\right) - \frac{\partial \Phi}{\partial \theta}\left(x, s_t^{\beta,\eta}, \theta_t^{\beta,\eta}\right) \right), \qquad (8)$$

which corresponds to the parameter gradient at time $t$, defined informally in Eq. (5). The following result makes this statement more formal.

**Theorem 1** (GDD Property). *Let $s_0, s_1, \ldots, s_T$ be the convergent sequence of states and denote $s_* = s_T$ the steady state. Further assume that there exists some step $K$ where $0 < K \leq T$ such that $s_* = s_T = s_{T-1} = \ldots s_{T-K}$. Then, in the limit $\eta \to 0$ and $\beta \to 0$, the first $K$ normalized updates in the second phase of C-EP are equal to the negatives of the first $K$ gradients of BPTT, i.e.*

$$\forall t = 0, 1, \ldots, K : \quad \left\{ \begin{array}{l} \lim\limits_{\beta \to 0} \lim\limits_{\eta \to 0} \Delta_s^{\mathrm{C-EP}}(\beta,\eta,t) = -\nabla_s^{\mathrm{BPTT}}(t), \\[2mm] \lim\limits_{\beta \to 0} \lim\limits_{\eta \to 0} \Delta_\theta^{\mathrm{C-EP}}(\beta,\eta,t) = -\nabla_\theta^{\mathrm{BPTT}}(t). \end{array} \right. \qquad (9)$$

Theorem 1 rewrites $s_{t+1}^{\beta,\eta} \approx s_t^{\beta,\eta} - \beta \frac{\partial \mathcal{L}}{\partial s_{T-t}}$ and $\theta_{t+1}^{\beta,\eta} \approx \theta_t^{\beta,\eta} - \eta \frac{\partial \mathcal{L}}{\partial \theta_{T-t}}$, showing that in the second phase of C-EP, neurons and synapses descend the gradients of the loss $\mathcal{L}$ obtained with BPTT, with the hyperparameters $\beta$ and $\eta$ playing the role of learning rates for $s_t^{\beta,\eta}$ and $\theta_t^{\beta,\eta}$, respectively. Fig. 3 illustrates Theorem 1 with a simple dynamical system for which the normalized updates $\Delta^{\mathrm{C-EP}}$ and the gradients $\nabla^{\mathrm{BPTT}}$ are analytically tractable - see Appendix C for derivation details.

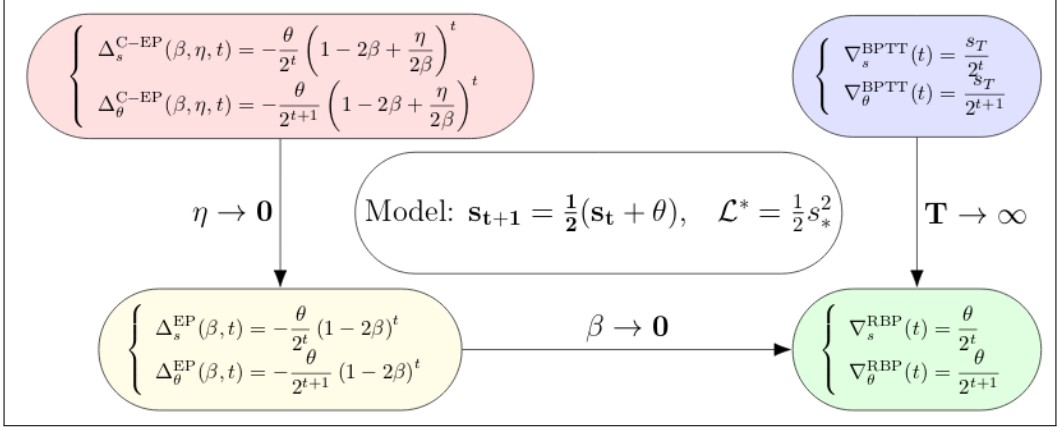

Figure 3: Illustration of Theorem 1 on a simple model. The variables $s$ and $\theta$ are scalars, the first phase equation is $s_{t+1} = \frac{1}{2}(s_t + \theta)$, the steady state is denoted $s_*$ and the loss is $\mathcal{L}^* = \frac{1}{2}s_*^2$. For completeness, we also include the corresponding gradients of Recurrent Backpropagation (RBP) and the normalized updates of EP, denoted $\nabla^{\text{RBP}}$ and $\Delta^{\text{EP}}$ respectively. The equivalence between C-EP, EP, RBP and BPTT holds in the general setting: see Appendix A for a thorough study of their relationship.

## 4 NUMERICAL EXPERIMENTS

In this section, we validate our continual version of Equilibrium Propagation against training on the MNIST data set with two models. The first model is a vanilla RNN with tied and symmetric weights: the dynamics of this model approximately derive from a primitive function, which allows training with C-EP. The second model is a Discrete-Time RNN with untied and asymmetric weights, which is therefore closer to biology. We train this second model with a modified version of C-EP which we call C-VF (*Continual Vector Field*) as it is inspired from the algorithm with Vector-Field dynamics of Scellier et al. (2018). Ernoult et al. (2019) showed with simulations the intuitive result that, if a model is such that the normalized updates of EP 'match' the gradients of BPTT (i.e. if they are approximately equal), then the model trained with EP performs as well as the model trained with BPTT. Along the same lines, we show in this work that the more the EP normalized updates follow the gradients of BPTT *before training*, the best is the resulting training performance. We choose to implement C-EP and C-VF on vanilla RNNs to accelerate simulations (Ernoult et al., 2019).

### 4.1 MODELS: VANILLA RNNS WITH SYMMETRIC AND ASYMMETRIC WEIGHTS

**Vanilla RNN with symmetric weights trained by C-EP.** The first phase dynamics is defined as:

$$s_{t+1} = \sigma\left(W \cdot s_t + W_x \cdot x\right), \tag{10}$$

where $\sigma$ is an activation function, $W$ is a symmetric weight matrix connecting the layers $s$ and $W_x$ is a matrix connecting the input $x$ to the layers $s$. Although the dynamics are not directly defined in terms of a primitive function, note that $s_{t+1} \approx \frac{\partial \Phi}{\partial s}(s_t, W)$ with $\Phi(s, W) = \frac{1}{2}s^\top \cdot W \cdot s$ if we ignore the activation function $\sigma$. Following Eq. (6) and Eq. (7), we define the normalized updates of this model as:

$$\Delta_s^{\text{C−EP}}(\beta, \eta, t) = \frac{1}{\beta}\left(s_{t+1}^{\beta,\eta} - s_t^{\beta,\eta}\right), \quad \Delta_W^{\text{C−EP}}(\beta, \eta, t) = \frac{1}{\beta}\left(s_{t+1}^{\beta,\eta^\top} \cdot s_{t+1}^{\beta,\eta} - s_t^{\beta,\eta^\top} \cdot s_t^{\beta,\eta}\right). \tag{11}$$

Note that this model applies to any topology as long as existing connections have symmetric values: this includes deep networks with any number of layers – see Appendix E for detailed descriptions of the models used in the experiments. More explicitly, for a network whose layers of neurons are $s^0, s^1, ..., s^N$, with $W_{n,n+1}$ connecting the layers $s^{n+1}$ and $s^n$ in both directions, the corresponding primitive function is $\Phi = \sum_n (s^n)^\top \cdot W_{n,n+1} \cdot s_{n+1} + s^{N^\top} \cdot W_x \cdot x$ - see Appendix E.1 for more details.

**Vanilla RNN with asymmetric weights trained by C-VF.** In this model, the dynamics in the first phase is the same as Eq. (10) but now the weight matrix $W$ is no longer assumed to be symmetric, i.e. the reciprocal connections between neurons are not constrained. In this setting the weight dynamics in the second phase is replaced by a version for asymmetric weights: $W_{t+1}^{\beta,\eta} = W_t^{\beta,\eta} + \frac{\eta}{\beta} s_t^{\beta,\eta^\top} \cdot \left( s_{t+1}^{\beta,\eta} - s_t^{\beta,\eta} \right)$, so that the normalized updates are equal to:

$$\Delta_s^{\mathrm{C-VF}}(\beta,\eta,t) = \frac{1}{\beta}\left(s_{t+1}^{\beta,\eta} - s_t^{\beta,\eta}\right), \qquad \Delta_W^{\mathrm{C-VF}}(\beta,\eta,t) = \frac{1}{\beta} s_t^{\beta,\eta^\top} \cdot \left(s_{t+1}^{\beta,\eta} - s_t^{\beta,\eta}\right). \quad (12)$$

Like the previous model, the vanilla RNN with asymmetric weights also applies to deep networks with any number of layers. Although in C-VF the dynamics of the weights is not one of the form of Eq. (6) that derives from a primitive function, the (bioplausible) normalized weight updates of Eq. (12) can approximately follow the gradients of BPTT, provided that the values of reciprocal connections are not too dissimilar: this is illustrated in Fig. 5 (as well as in Fig. 12 and Fig. 13 of Appendix E.6) and proved in Appendix D.2. This property motivates the following training experiments.

| | Error (%) | | T | K | Epochs |
|---|---|---|---|---|---|
| | Test | Train | | | |
| EP-1h | $2.00 \pm 0.13$ | $(0.20)$ | 30 | 10 | 30 |
| EP-2h | $1.95 \pm 0.10$ | $(0.14)$ | 100 | 20 | 50 |
| C-EP-1h | $\mathbf{2.28 \pm 0.16}$ | $(0.41)$ | 40 | 15 | 100 |
| C-EP-2h | $\mathbf{2.44 \pm 0.14}$ | $(0.31)$ | 100 | 20 | 150 |
| C-VF-1h | $\mathbf{2.43 \pm 0.08}$ | $(0.77)$ | 40 | 15 | 100 |
| C-VF-2h | $\mathbf{2.97 \pm 0.19}$ | $(1.58)$ | 100 | 20 | 150 |

Figure 4: **Left:** Training results on MNIST with EP, C-EP and C-VF. "#h" stands for the number of hidden layers. We indicate over 5 trials the mean and standard deviation for the test error (mean train error in parenthesis). $T$ (resp. $K$) is the number of iterations in the 1$^{\mathrm{st}}$ (resp. 2$^{\mathrm{nd}}$) phase. For C-VF results, the initial angle between forward ($\theta_\mathrm{f}$) and backward ($\theta_\mathrm{b}$) weights is $\Psi(\theta_\mathrm{f}, \theta_\mathrm{b}) = 0°$. **Right:** Test error rate on MNIST achieved by C-VF as a function of the initial $\Psi(\theta_\mathrm{f}, \theta_\mathrm{b})$.

## 4.2 C-EP TRAINING EXPERIMENTS

Experiments are performed with multi-layered vanilla RNNs (with symmetric weights) on MNIST. The table of Fig. 4.1 presents the results obtained with C-EP training benchmarked against standard EP training (Ernoult et al., 2019) - see Appendix E for model details and Appendix F.1 for training conditions. Although the test error of C-EP approaches that of EP, we observe a degradation in accuracy. This is because although Theorem 1 guarantees Gradient Descending Dynamics (GDD) in the limit of infinitely small learning rates, in practice we have to strike a balance between having a learning rate that is small enough to ensure this condition but not too small to observe convergence within a reasonable number of epochs. As seen on Fig. 5 (b), the finite learning rate $\eta$ of continual updates leads to $\Delta^{\mathrm{C-EP}}(\beta,\eta,t)$ curves splitting apart from the $-\nabla^{\mathrm{BPTT}}(t)$ curves. As seen per Fig. 5 (a), this effect is emphasized with the depth: before training, angles between the normalized updates of C-EP and the gradients of BPTT reach 50 degrees for two hidden layers. The deeper the network, the more difficult it is for the C-EP dynamics to follow the gradients provided by BPTT. As an evidence, we show in Appendix F.2 that when we use extremely small learning rates throughout the second phase ($\theta \leftarrow \theta + \eta_{\mathrm{tiny}}\Delta_\theta^{\mathrm{C-EP}}$) and rescale up the resulting total weight update ($\theta \leftarrow \theta - \Delta\theta_{\mathrm{tot}} + \frac{\eta}{\eta_{\mathrm{tiny}}}\Delta\theta_{\mathrm{tot}}$), we recover standard EP results.

## 4.3 CONTINUAL VECTOR FIELD (C-VF) TRAINING EXPERIMENTS

Depending on whether the updates occur continuously during the second phase and the system obey general dynamics with untied forward and backward weights, we can span a large range of deviations from the ideal conditions of Theorem 1. Fig. 5 (b) qualitatively depicts these deviations with a model for which the normalized updates of EP match the gradients of BPTT (EP) ; with continual weight

updates, the normalized updates and gradients start splitting apart (C-EP), and even more so if the weights are untied (C-VF).

**Protocol.** In order to create these deviations from Theorem 1 and study the consequences in terms of training, we proceed as follows. For each C-VF simulations, we tune the initial angle between forward weights ($\theta_f$) and backward weights ($\theta_b$) between 0 and $180°$. We denote this angle $\Psi(\theta_f, \theta_b)$ - see Appendix F.1 for the angle definition and the angle tuning technique employed. For each of these weight initialization, we compute the angle between the *total* normalized update provided by C-VF, i.e. $\Delta^{C-VF}(\beta, \eta, \text{tot}) = \sum_{t=0}^{K-1} \Delta^{C-VF}(\beta, \eta, t)$ and the *total* gradient provided by BPTT, i.e. $\nabla^{BPTT}(\text{tot}) = \sum_{t=0}^{K-1} \nabla^{BPTT}(t)$ on random mini-batches *before training*. We denote this angle $\Psi\left(\Delta^{C-VF}(\text{tot}), -\nabla^{BPTT}(\text{tot})\right)$. Finally for each weight initialization, we perform training in the discrete-time setting of Ernoult et al. (2019) - see Appendix E.4 for model details. We proceed in the same way for EP and C-EP simulations, computing $\Psi\left(\Delta^{EP}(\text{tot}), -\nabla^{BPTT}(\text{tot})\right)$ and $\Psi\left(\Delta^{C-EP}(\text{tot}), -\nabla^{BPTT}(\text{tot})\right)$ before training. We use the generic notation $\Delta(\text{tot})$ to denote the total normalized update. This procedure yields $(x, y)$ data points with $x = \Psi\left(\Delta(\text{tot}), -\nabla^{BPTT}(\text{tot})\right)$ and $y = $ test error, which are reported on Fig. 5 (a) - see Appendix F.1 for the full table of results.

**Results.** Fig. 5 (a) shows the test error achieved on MNIST by EP, C-EP on the vanilla RNN with symmetric weights and C-VF on the vanilla RNN with asymmetric weights for different number of hidden layers as a function of the angle $\Psi\left(\Delta(\text{tot}), -\nabla^{BPTT}(\text{tot})\right)$ before training. This graphical representation spreads the algorithms between EP which best satisfies the GDD property (leftmost point in green at $\sim 20°$) to C-VF which satisfies the less the GDD property (rightmost points in red and orange at $\sim 100°$). As expected, high angles between gradients of C-VF and BPTT lead to high error rates that can reach $90\%$ for $\Psi\left(\Delta^{C-VF}(\text{tot}), -\nabla^{BPTT}(\text{tot})\right)$ over $100°$. More precisely, the inset of Fig. 5 shows the same data but focusing only on results generated by initial weight angles lying below $90°$, i.e. $\Psi(\theta_f, \theta_b) = \{0°, 22.5°, 45°, 67.5°, 90°\}$. From standard EP with one hidden layer to C-VF with two hidden layers, the test error increases monotonically with $\Psi\left(\Delta(\text{tot}), -\nabla^{BPTT}(\text{tot})\right)$ but does not exceed $5.05\%$ on average. This result confirms the importance of proper weight initialization when weights are untied, also discussed in other context (Lillicrap et al., 2016). When the initial weight angle is of $0°$, the impact of untying the weights on classification accuracy remains constrained, as shown in table of Fig. 4.1. Upon untying the forward and backward weights, the test error increases by $\sim 0.2\%$ with one hidden layer and by $\sim 0.5\%$ with two hidden layers compared to standard C-EP.

## 5 DISCUSSION

Equilibrium Propagation is an algorithm that leverages the dynamical nature of neurons to compute weight gradients through the physics of the neural network. C-EP embraces simultaneous synapse and neuron dynamics, resolving the initial need of artificial memory units for storing the neuron values between different phases. The C-EP framework preserves the equivalence with Backpropagation Through Time: in the limit of sufficiently slow synaptic dynamics (i.e. small learning rates), the system satisfies Gradient Descending Dynamics (Theorem 1). Our experimental results confirm this theorem. When training our vanilla RNN with symmetric weights with C-EP while ensuring convergence in 100 epochs, a modest reduction in MNIST accuracy is seen with regards to standard EP. This accuracy reduction can be eliminated by using smaller learning rates and rescaling up the total weight update at the end of the second phase (Appendix F.2). On top of extending the theory of Ernoult et al. (2019), Theorem 1 also appears to provide a statistically robust tool for C-EP based learning. Our experimental results show as in Ernoult et al. (2019) that, for a given network with specified neuron and synapse dynamics, the more the updates of Equilibrium Propagation follow the gradients provided by Backpropagation Through Time before training (in terms of angle in this work), the better this network can learn.

Our C-EP and C-VF algorithms exhibit features reminiscent of biology. C-VF extends C-EP training to RNNs with asymmetric weights between neurons, as is the case in biology. Its learning rule, local in space and time, is furthermore closely acquainted to Spike Timing Dependent Plasticity (STDP), a learning rule widely studied in neuroscience, inferred in vitro and in vivo from neural recordings in the hippocampus (Dan and Poo, 2004). In STDP, the synaptic strength is modulated by the relative timings of pre and post synaptic spikes within a precise time window (Bi and Poo, 1998; 2001).

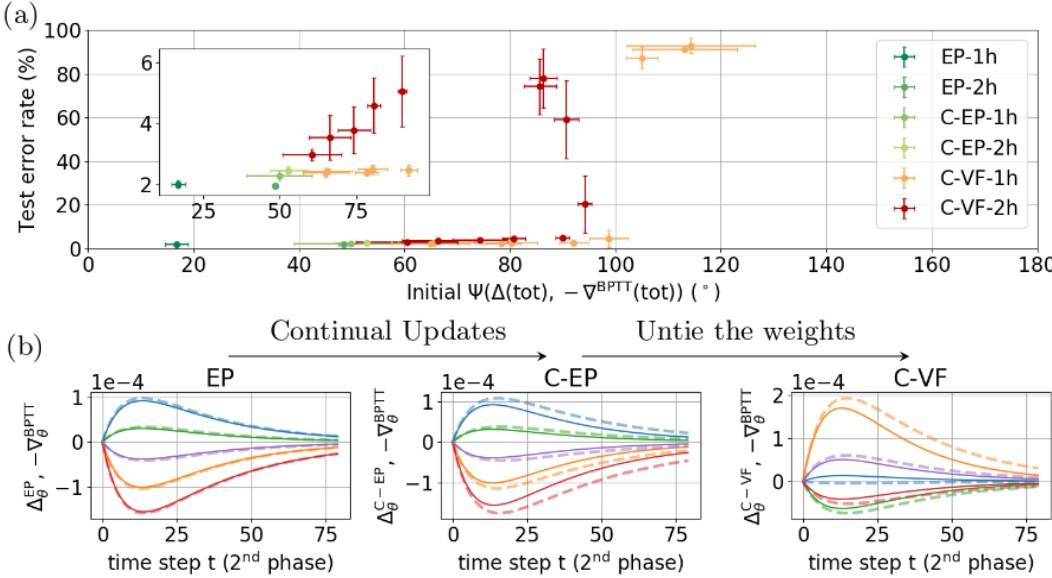

Figure 5: **Three versions of EP:** standard Equilibrium Propagation (EP), Continual Equilibrium Propagation (C-EP) and Continual Vector Field EP (C-VF). #-h denotes the number of hidden layers. **(a)**: test error rate on MNIST as a function of the initial angle $\Psi$ between the total normalized update of EP and the total gradient of BPTT. **(b)**: Dashed and continuous lines respectively represent the normalized updates $\Delta_\theta(t)$ (i.e. $\Delta_\theta^{\mathrm{EP}}(t)$, $\Delta_\theta^{\mathrm{C-EP}}(t)$, $\Delta_\theta^{\mathrm{C-VF}}(t)$) and the gradients $-\nabla_\theta^{\mathrm{BPTT}}(t)$. Each randomly selected synapse corresponds to one color. While dashed and continuous lines coincide for standard EP, they split apart upon untying the weights and using continual updates.

Strikingly, the same rule that we use for C-VF learning can approximate STDP correlations in a rate-based formulation, as shown through numerical experiments by Bengio et al. (2015). From this viewpoint our work brings EP a step closer to biology. However, C-EP and C-VF do not aim at being models of biological learning per se, in that it would account for how the brain works or how animals learn, for which Reinforcement Learning might be a more suited learning paradigm. The core motivation of this work is to propose a fully local implementation of EP, in particular to foster its hardware implementation.

When computed on a standard computer, due to the use of small learning rates to mimic analog dynamics within a finite number of epochs, training our models with C-EP and C-VF entail long simulation times. With a Titan RTX GPU, training a fully connected architecture on MNIST takes 2 hours 39 mins with 1 hidden layer and 10 hours 49 mins with 2 hidden layers. On the other hand, C-EP and C-VF might be particularly efficient in terms of speed and energy consumption when operated on neuromorphic hardware that employs analog device physics (Ambrogio et al., 2018; Romera et al., 2018). To this purpose, our work can provide an engineering guidance to map our algorithm onto a neuromorphic system. Fig. 5 (a) shows that hyperparameters should be tuned so that before training, C-EP updates stay within $90°$ of the gradients provided by BPTT. More concretely in practice, it amounts to tune the degree of symmetry of the dynamics, for instance the angle between forward and backward weights - see Fig. 4.1. Our work is one step towards bridging Equilibrium Propagation with neuromorphic computing and thereby energy efficient implementations of gradient-based learning algorithms.

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

# Appendix

## A  PROOF OF THEOREM 1

In this appendix, we prove Theorem 1, which we recall here.

**Theorem 1** (GDD Property). *Let $s_0, s_1, \ldots, s_T$ be the convergent sequence of states and denote $s_* = s_T$ the steady state. Further assume that there exists some step $K$ where $0 < K \leq T$ such that $s_* = s_T = s_{T-1} = \ldots s_{T-K}$. Then, in the limit $\eta \to 0$ and $\beta \to 0$, the first $K$ normalized updates in the second phase of C-EP are equal to the negatives of the first $K$ gradients of BPTT, i.e.*

$$\forall t = 0, 1, \ldots, K : \begin{cases} \lim_{\beta \to 0} \lim_{\eta \to 0} \Delta_s^{C-EP}(\beta, \eta, t) = -\nabla_s^{BPTT}(t), \\ \lim_{\beta \to 0} \lim_{\eta \to 0} \Delta_\theta^{C-EP}(\beta, \eta, t) = -\nabla_\theta^{BPTT}(t). \end{cases} \tag{9}$$

### A.1  A SPECTRUM OF FOUR COMPUTATIONALLY EQUIVALENT LEARNING ALGORITHMS

Proving Theorem 1 amounts to prove the equivalence of C-EP and BPTT. In fact we can prove the equivalence of four algorithms, which all compute the gradient of the loss:

1. Backpropagation Through Time (BPTT), presented in Section B.2,
2. Recurrent Backpropagation (RBP), presented in Section B.3,
3. Equilibrium Propagation (EP), presented in Section 2,
4. Equilibrium Propagation with Continual Weight Updates (C-EP), introduced in Section 3.

In this spectrum of algorithms, BPTT is the most practical algorithm to date from the point of view of machine learning, but also the less biologically realistic. In contrast, C-EP is the most realistic in terms of implementation in biological systems, while it is to date the least practical and least efficient for conventional machine learning (computations on standard Von-Neumann hardware are considerably slower due to repeated parameter updates, requiring memory access at each time-step of the second phase).

### A.2  SKETCH OF THE PROOF

Theorem 1 can be proved in three phases, using the following three lemmas.

**Lemma 2** (Equivalence of C-EP and EP). *In the limit of small learning rate, i.e. $\eta \to 0$, the (normalized) updates of C-EP are equal to those of EP:*

$$\forall t \geq 0 : \begin{cases} \lim_{\eta \to 0 \ (\eta > 0)} \Delta_s^{C-EP}(\beta, \eta, t) = \Delta_s^{EP}(\beta, t), \\ \lim_{\eta \to 0 \ (\eta > 0)} \Delta_\theta^{C-EP}(\beta, \eta, t) = \Delta_\theta^{EP}(\beta, t). \end{cases} \tag{13}$$

**Lemma 3** (Equivalence of EP and RBP). *Assume that the transition function derives from a primitive function, i.e. that $F$ is of the form $F(x, s, \theta) = \frac{\partial \Phi}{\partial s}(x, s, \theta)$. Then, in the limit of small hyperparameter $\beta$, the normalized updates of EP are equal to the gradients of RBP:*

$$\forall t \geq 0 : \begin{cases} \lim_{\beta \to 0 \ (\beta > 0)} \Delta_s^{EP}(\beta, t) = -\nabla_s^{RBP}(t), \\ \lim_{\beta \to 0 \ (\beta > 0)} \Delta_\theta^{EP}(\beta, t) = -\nabla_\theta^{RBP}(t). \end{cases} \tag{14}$$

**Lemma 4** (Equivalence of BPTT and RBP). *In the setting with static input $x$, suppose that the network has reached the steady state $s_*$ after $T - K$ steps, i.e. $s_{T-K} = s_{T-K+1} = \cdots = s_{T-1} = s_T = s_*$. Then the first $K$ gradients of BPTT are equal to the first $K$ gradient of RBP, i.e.*

$$\forall t = 0, 1, \ldots, K : \begin{cases} \nabla_s^{BPTT}(t) = \nabla_s^{RBP}(t), \\ \nabla_\theta^{BPTT}(t) = \nabla_\theta^{RBP}(t). \end{cases} \tag{15}$$

Proofs of the Lemmas can be found in the following places:

- The link between BPTT and RBP (Lemma 2) is known since the late 1980s and can be found e.g. in Hertz (2018). We also prove it here in Appendix B.
- Lemma 3 was proved in Scellier and Bengio (2019) in the setting of real-time dynamics.
- Lemma 4 is the new ingredient contributed here, and we prove it in Appendix A.3.

Also a direct proof of the equivalence of EP and BPTT was derived in Ernoult et al. (2019).

## A.3 EQUIVALENCE OF C-EP AND EP

First, recall the dynamics of C-EP in the second phase: starting from $s_0^{\beta,\eta} = s_*$ and $\theta_0^{\beta,\eta} = \theta$ we have

$$\forall t \geq 0 : \begin{cases} s_{t+1}^{\beta,\eta} = \dfrac{\partial \Phi}{\partial s}\left(x, s_t^{\beta,\eta}, \theta_t^{\beta,\eta}\right) - \beta\,\dfrac{\partial \ell}{\partial s}\left(s_t^{\beta,\eta}, y\right), \\[2mm] \theta_{t+1}^{\beta,\eta} = \theta_t^{\beta,\eta} + \dfrac{\eta}{\beta}\left(\dfrac{\partial \Phi}{\partial \theta}\left(x, s_{t+1}^{\beta,\eta}, \theta_t^{\beta,\eta}\right) - \dfrac{\partial \Phi}{\partial \theta}\left(x, s_t^{\beta,\eta}, \theta_t^{\beta,\eta}\right)\right). \end{cases} \tag{16}$$

We have also defined the normalized updates of C-EP:

$$\forall t \geq 0 : \begin{cases} \Delta_s^{C-EP}(\beta, \eta, t) = \dfrac{1}{\beta}\left(s_{t+1}^{\beta,\eta} - s_t^{\beta,\eta}\right), \\[2mm] \Delta_\theta^{C-EP}(\beta, \eta, t) = \dfrac{1}{\eta}\left(\theta_{t+1}^{\beta,\eta} - \theta_t^{\beta,\eta}\right). \end{cases} \tag{17}$$

We also recall the dynamics of EP in the second phase:

$$s_0^\beta = s_* \quad \text{and} \quad s_{t+1}^\beta = \dfrac{\partial \Phi}{\partial s}\left(x, s_t^\beta, \theta\right) - \beta\,\dfrac{\partial \ell}{\partial s}\left(s_t^\beta, y\right), \tag{18}$$

as well as the normalized updates of EP, as defined in Ernoult et al. (2019):

$$\forall t \geq 0 : \begin{cases} \Delta_s^{EP}(\beta, t) = \dfrac{1}{\beta}\left(s_{t+1}^\beta - s_t^\beta\right), \\[2mm] \Delta_\theta^{EP}(\beta, t) = \dfrac{1}{\beta}\left(\dfrac{\partial \Phi}{\partial \theta}\left(x, s_{t+1}^\beta, \theta\right) - \dfrac{\partial \Phi}{\partial \theta}\left(x, s_t^\beta, \theta\right)\right). \end{cases} \tag{19}$$

**Lemma 2** (Equivalence of C-EP and EP)**.** *In the limit of small learning rate, i.e. $\eta \to 0$, the (normalized) updates of C-EP are equal to those of EP:*

$$\forall t \geq 0 : \begin{cases} \lim\limits_{\eta \to 0\ (\eta > 0)} \Delta_s^{C-EP}(\beta, \eta, t) = \Delta_s^{EP}(\beta, t), \\[2mm] \lim\limits_{\eta \to 0\ (\eta > 0)} \Delta_\theta^{C-EP}(\beta, \eta, t) = \Delta_\theta^{EP}(\beta, t). \end{cases} \tag{13}$$

*Proof of Lemma 2.* We want to compute the limits of $\Delta_s^{C-EP}(\beta, \eta, t)$ and $\Delta_\theta^{C-EP}(\beta, \eta, t)$ as $\eta \to 0$ with $\eta > 0$. First of all, note that under mild assumptions – which we made here – of regularity on the functions $\Phi$ and $\ell$ (e.g. continuous differentiability), for fixed $t$ and $\beta$, the quantities $s_t^{\beta,\eta}$ and $\theta_t^{\beta,\eta}$ are continuous as functions of $\eta$ ; this is straightforward from the form of Eq. (16). As a consequence, $\Delta_s^{C-EP}(\beta, \eta, t)$ is a continuous function of $\eta$, which implies in particular that

$$\lim_{\eta \to 0\ (\eta > 0)} \Delta_s^{C-EP}(\beta, \eta, t) = \Delta_s^{C-EP}(\beta, 0, t). \tag{20}$$

Now, taking $\eta = 0$ in the bottom equation of Eq. (16) yields the recurrence relation $\theta_{t+1}^{\beta,0} = \theta_t^{\beta,0}$, so that $\theta_t^{\beta,0} = \theta_0^{\beta,0} = \theta$ for every $t$. Injecting $\theta_t^{\beta,0} = \theta$ in the top equation of Eq. (16) yields for $s_t^{\beta,0}$ the same recurrence relation as that of $s_t^\beta$ (Eq. 18), so that $s_t^{\beta,0} = s_t^\beta$ for every $t$. Therefore, for $\eta = 0$, we have

$$\Delta_s^{C-EP}(\beta, 0, t) = \dfrac{1}{\beta}\left(s_{t+1}^{\beta,0} - s_t^{\beta,0}\right) = \dfrac{1}{\beta}\left(s_{t+1}^\beta - s_t^\beta\right) = \Delta_s^{EP}(\beta, t). \tag{21}$$

It follows from Eq. (20) and Eq. (21) that

$$\lim_{\eta \to 0\ (\eta > 0)} \Delta_s^{C-EP}(\beta, \eta, t) = \Delta_s^{EP}(\beta, t). \tag{22}$$

Now let us compute $\lim_{\eta \to 0 \ (\eta > 0)} \Delta_s^{\mathrm{C-EP}}(\beta, \eta, t)$. Using Eq. (16), we have

$$\Delta_\theta^{\mathrm{C-EP}}(\beta, \eta, t) = \frac{1}{\eta} \left( \theta_{t+1}^{\beta,\eta} - \theta_t^{\beta,\eta} \right) \tag{23}$$

$$= \frac{1}{\beta} \left( \frac{\partial \Phi}{\partial \theta} \left( x, s_{t+1}^{\beta,\eta}, \theta_t^{\beta,\eta} \right) - \frac{\partial \Phi}{\partial \theta} \left( x, s_t^{\beta,\eta}, \theta_t^{\beta,\eta} \right) \right). \tag{24}$$

Similarly as before, for fixed $t$, $\frac{\partial \Phi}{\partial \theta} \left( x, s_t^{\beta,\eta}, \theta_t^{\beta,\eta} \right)$ is a continuous function of $\eta$. Therefore

$$\lim_{\eta \to 0 \ (\eta > 0)} \Delta_\theta^{\mathrm{C-EP}}(\beta, \eta, t) = \frac{1}{\beta} \left( \frac{\partial \Phi}{\partial \theta} \left( x, s_{t+1}^{\beta,0}, \theta_t^{\beta,0} \right) - \frac{\partial \Phi}{\partial \theta} \left( x, s_t^{\beta,0}, \theta_t^{\beta,0} \right) \right) \tag{25}$$

$$= \frac{1}{\beta} \left( \frac{\partial \Phi}{\partial \theta} \left( x, s_{t+1}^{\beta}, \theta \right) - \frac{\partial \Phi}{\partial \theta} \left( x, s_t^{\beta}, \theta \right) \right) = \Delta_\theta^{\mathrm{EP}}(\beta, t). \tag{26}$$

$\square$

A consequence of Lemma 2 is that the total update of C-EP matches the total update of EP in the limit of small $\eta$, so that we retrieve the standard EP learning rule of Eq. (4). More explicitly, after K steps in the second phase and starting from $\theta_0^{\beta,\eta} = \theta_0$:

$$\theta_K^{\beta,\eta} - \theta_0 = \sum_{t=0}^{K-1} \theta_{t+1}^{\beta,\eta} - \theta_t^{\beta,\eta}$$

$$= \sum_{t=0}^{K-1} \eta \Delta_\theta^{\mathrm{C-EP}}(\beta, \eta, t) \qquad \text{(definition of } \Delta_\theta^{\mathrm{C-EP}}, \text{ Eq. (7))}$$

$$\underset{\eta \gtrsim 0}{\approx} \sum_{t=0}^{K-1} \eta \Delta_\theta^{\mathrm{EP}}(\beta, t) \qquad \text{(Lemma 2)}$$

$$= \sum_{t=0}^{K-1} \eta \frac{1}{\beta} \left( \frac{\partial \Phi}{\partial \theta}(x, s_{t+1}, \theta_0) - \frac{\partial \Phi}{\partial \theta}(x, s_{t+1}, \theta_0) \right) \qquad \text{(definition of } \Delta_\theta^{\mathrm{EP}}, \text{ Eq. (19))}$$

$$= \frac{\eta}{\beta} \left( \frac{\partial \Phi}{\partial \theta}(x, s_K^{\beta}, \theta_0) - \frac{\partial \Phi}{\partial \theta}(x, s_*, \theta_0) \right)$$

## B  EQUIVALENCE OF BPTT AND RBP

In this section, we recall Backprop Through Time (BPTT) and the Almeida-Pineda Recurrent Backprop (RBP) algorithm, which can both be used to optimize the loss $\mathcal{L}^*$ of Eq. 3. Historically, BPTT and RBP were invented separately around the same time. RBP was introduced at a time when convergent RNNs (such as the one studied in this paper) were popular. Nowadays, convergent RNNs are less popular ; in the field of deep learning, RNNs are almost exclusively used for tasks that deal with sequential data and BPTT is the algorithm of choice to train such RNNs. Here, we present RBP in a way that it can be seen as a particular case of BPTT.

### B.1  SKETCH OF THE PROOF OF LEMMA 4

Lemma 4, which we recall here, is a consequence of Proposition 5 and Definition 6 below.

**Lemma 4** (Equivalence of BPTT and RBP). *In the setting with static input $x$, suppose that the network has reached the steady state $s_*$ after $T - K$ steps, i.e. $s_{T-K} = s_{T-K+1} = \cdots = s_{T-1} = s_T = s_*$. Then the first $K$ gradients of BPTT are equal to the first $K$ gradient of RBP, i.e.*

$$\forall t = 0, 1, \ldots, K : \left\{ \begin{array}{l} \nabla_s^{\mathrm{BPTT}}(t) = \nabla_s^{\mathrm{RBP}}(t), \\ \nabla_\theta^{\mathrm{BPTT}}(t) = \nabla_\theta^{\mathrm{RBP}}(t). \end{array} \right. \tag{15}$$

However, in general, the gradients $\nabla^{\mathrm{BPTT}}(t)$ of BPTT and the gradients $\nabla^{\mathrm{RBP}}(t)$ of RBP are not equal for $t > K$. This is because BPTT and RBP compute the gradients of different loss functions:

- BPTT computes the gradient of the loss after $T$ time steps, i.e. $\mathcal{L} = \ell(s_T, y)$,

- RBP computes the gradients of the loss at the steady state, i.e. $\mathcal{L}^* = \ell(s_*, y)$.

## B.2 BACKPROPAGATION THROUGH TIME (BPTT)

Backpropagation Through Time (BPTT) is the standard method to train RNNs and can also be used to train the kind of convergent RNNs that we study in this paper. To this end, we consider the cost of the state $s_T$ after $T$ time steps, denoted $\mathcal{L} = \ell(s_T, y)$, and we substitute the loss after $T$ time steps $\mathcal{L}$ as a proxy for the loss at the steady state $\mathcal{L}^* = \ell(s_*, y)$. The gradients of $\mathcal{L}$ can then be computed with BPTT.

To do this, we recall some of the inner working mechanisms of BPTT. Eq. (1) rewrites in the form $s_{t+1} = F(x, s_t, \theta_{t+1} = \theta)$, where $\theta_t$ denotes the parameter of the model at time step $t$, the value $\theta$ being shared across all time steps. This way of rewriting Eq. (1) enables us to define the partial derivative $\frac{\partial \mathcal{L}}{\partial \theta_t}$ as the sensitivity of the loss $\mathcal{L}$ with respect to $\theta_t$ when $\theta_1, \dots \theta_{t-1}, \theta_{t+1}, \dots \theta_T$ remain fixed (set to the value $\theta$). With these notations, the gradient $\frac{\partial \mathcal{L}}{\partial \theta}$ reads as the sum:

$$\frac{\partial \mathcal{L}}{\partial \theta} = \frac{\partial \mathcal{L}}{\partial \theta_1} + \frac{\partial \mathcal{L}}{\partial \theta_2} + \dots + \frac{\partial \mathcal{L}}{\partial \theta_T}. \tag{27}$$

BPTT computes the 'full' gradient $\frac{\partial \mathcal{L}}{\partial \theta}$ by first computing the partial derivatives $\frac{\partial \mathcal{L}}{\partial s_t}$ and $\frac{\partial \mathcal{L}}{\partial \theta_t}$ iteratively, backward in time, using the chain rule of differentiation. In this work, we denote the gradients that BPTT computes:

$$\forall t \in [0, T-1] : \begin{cases} \nabla_s^{\text{BPTT}}(t) = \dfrac{\partial \mathcal{L}}{\partial s_{T-t}}, \\ \nabla_\theta^{\text{BPTT}}(t) = \dfrac{\partial \mathcal{L}}{\partial \theta_{T-t}}. \end{cases} \tag{28}$$

**Proposition 5** (Gradients of BPTT). *The gradients $\nabla_s^{\text{BPTT}}(t)$ and $\nabla_\theta^{\text{BPTT}}(t)$ satisfy the recurrence relationship*

$$\nabla_s^{\text{BPTT}}(0) = \frac{\partial \ell}{\partial s}(s_T, y), \tag{29}$$

$$\forall t = 1, 2, \dots, T, \qquad \nabla_s^{\text{BPTT}}(t) = \frac{\partial F}{\partial s}(x, s_{T-t}, \theta)^\top \cdot \nabla_s^{\text{BPTT}}(t-1), \tag{30}$$

$$\forall t = 1, 2, \dots, T, \qquad \nabla_\theta^{\text{BPTT}}(t) = \frac{\partial F}{\partial \theta}(x, s_{T-t}, \theta)^\top \cdot \nabla_s^{\text{BPTT}}(t-1). \tag{31}$$

## B.3 FROM BACKPROP THROUGH TIME (BPTT) TO RECURRENT BACKPROP (RBP)

In general, to apply BPTT, it is necessary to store in memory the history of past hidden states $s_1, s_2, \dots, s_T$ in order to compute the gradients $\nabla_s^{\text{BPTT}}(t)$ and $\nabla_\theta^{\text{BPTT}}(t)$ as in Eq. 30-31. However, in our specific setting with static input $x$, if the network has reached the steady state $s_*$ after $T - K$ steps, i.e. if $s_{T-K} = s_{T-K+1} = \dots = s_{T-1} = s_T = s_*$, then we see that, in order to compute the first $K$ gradients of BPTT, all one needs to know is $\frac{\partial F}{\partial s}(x, s_*, \theta)$ and $\frac{\partial F}{\partial \theta}(x, s_*, \theta)$. To this end, all one needs to keep in memory is the steady state $s_*$. In this particular setting, it is not necessary to store the past hidden states $s_T, s_{T-1}, \dots, s_{T-K}$ since they are all equal to $s_*$.

The Almeida-Pineda algorithm (a.k.a. Recurrent Backpropagation, or RBP for short), which was invented independently by Almeida (1987) and Pineda (1987), relies on this property to compute the gradients of the loss $\mathcal{L}^*$ using only the steady state $s_*$. Similarly to BPTT, it computes quantities $\nabla_s^{\text{RBP}}(t)$ and $\nabla_\theta^{\text{RBP}}(t)$, which we call 'gradients of RBP', iteratively for $t = 0, 1, 2, \dots$

**Definition 6** (Gradients of RBP). *The gradients $\nabla_s^{\mathrm{RBP}}(t)$ and $\nabla_\theta^{\mathrm{RBP}}(t)$ are defined and computed iteratively as follows:*

$$\nabla_s^{\mathrm{RBP}}(0) = \frac{\partial \ell}{\partial s}(s_*, y), \tag{32}$$

$$\forall t \geq 0, \qquad \nabla_s^{\mathrm{RBP}}(t+1) = \frac{\partial F}{\partial s}(x, s_*, \theta)^\top \cdot \nabla_s^{\mathrm{RBP}}(t), \tag{33}$$

$$\forall t \geq 0, \qquad \nabla_\theta^{\mathrm{RBP}}(t+1) = \frac{\partial F}{\partial \theta}(x, s_*, \theta)^\top \cdot \nabla_s^{\mathrm{RBP}}(t). \tag{34}$$

Unlike in BPTT where keeping the history of past hidden states is necessary to compute (or 'back-propagate') the gradients, in RBP Eq. 33-34 show that it is sufficient to keep in memory the steady state $s_*$ only in order to iterate the computation of the gradients. RBP is more memory efficient than BPTT.

---

**Algorithm 3** BPTT

*Input*: $x$, $y$, $\theta$.
*Output*: $\theta$.

1: $s_0 \leftarrow 0$
2: **for** $t = 0$ to $T - 1$ **do**
3:     $s_{t+1} \leftarrow F(x, s_t, \theta)$
4: **end for**
5: $\nabla_s^{\mathrm{BPTT}}(0) \leftarrow \frac{\partial \ell}{\partial s}(s_T, y)$
6: **for** $t = 1$ to $T$ **do**
7:     $\nabla_s(t) \leftarrow \frac{\partial F}{\partial s}(x, s_{T-t}, \theta)^\top \cdot \nabla_s(t-1)$
8:     $\nabla_\theta(t) \leftarrow \frac{\partial F}{\partial \theta}(x, s_{T-t}, \theta)^\top \cdot \nabla_s(t-1)$
9: **end for**
10: $\nabla_\theta^{\mathrm{BPTT}}(\mathrm{tot}) \leftarrow \sum_{t=0}^{T-1} \nabla_\theta^{\mathrm{BPTT}}(t)$

**Algorithm 4** RBP

*Input*: $x$, $y$, $\theta$.
*Output*: $\theta$.

1: $s_0 \leftarrow 0$
2: **repeat**
3:     $s_{t+1} \leftarrow F(x, s_t, \theta)$
4: **until** $s_t = s_*$
5: $\nabla_s^{\mathrm{RBP}}(0) \leftarrow \frac{\partial \ell}{\partial s}(s_*, y)$
6: **repeat**
7:     $\nabla_s^{\mathrm{RBP}}(t) \leftarrow \frac{\partial F}{\partial s}(x, s_*, \theta)^\top \cdot \nabla_s^{\mathrm{RBP}}(t-1)$
8:     $\nabla_\theta^{\mathrm{RBP}}(t) \leftarrow \frac{\partial F}{\partial \theta}(x, s_*, \theta)^\top \cdot \nabla_s^{\mathrm{RBP}}(t-1)$
9: **until** $\nabla_\theta^{\mathrm{RBP}}(t) = 0$.
10: $\nabla_\theta^{\mathrm{RBP}}(\mathrm{tot}) \leftarrow \sum_{t=0}^{\infty} \nabla_\theta^{\mathrm{RBP}}(t)$

---

Figure 6: **Left.** Pseudo-code of BPTT. The gradients $\nabla(t)$ denote the gradients $\nabla^{\mathrm{BPTT}}(t)$ of BPTT. **Right.** Pseudo-code of RBP. **Difference between BPTT and RBP.** In BPTT, the state $s_{T-t}$ is required to compute $\frac{\partial F}{\partial s}(x, s_{T-t}, \theta)$ and $\frac{\partial F}{\partial \theta}(x, s_{T-t}, \theta)$ ; thus it is necessary to store in memory the sequence of states $s_1, s_2, \ldots, s_T$. In contrast, in RBP, only the steady state $s_*$ is required to compute $\frac{\partial F}{\partial s}(x, s_*, \theta)$ and $\frac{\partial F}{\partial \theta}(x, s_*, \theta)$ ; it is not necessary to store the past states of the network.

### B.4 What 'Gradients' are the Gradients of RBP?

In this subsection we motivate the name of 'gradients' for the quantities $\nabla_s^{\mathrm{RBP}}(t)$ and $\nabla_\theta^{\mathrm{RBP}}(t)$ by proving that they are the gradients of $\mathcal{L}^*$ in the sense of Proposition 7 below. They are also the gradients of what we call the 'projected cost function' (Proposition 8), using the terminology of Scellier and Bengio (2019).

**Proposition 7** (RBP Optimizes $\mathcal{L}^*$). *The total gradient computed by the RBP algorithm is the gradient of the loss $\mathcal{L}^* = \ell(s_*, y)$, i.e.*

$$\sum_{t=1}^{\infty} \nabla_\theta^{\mathrm{RBP}}(t) = \frac{\partial \mathcal{L}^*}{\partial \theta}. \tag{35}$$

$\nabla_s^{\mathrm{RBP}}(t)$ and $\nabla_\theta^{\mathrm{RBP}}(t)$ can also be expressed as gradients of $\mathcal{L}_t = \ell(s_t, y)$, the cost after $t$ time steps. In the terminology of Scellier and Bengio (2019), $\mathcal{L}_t$ was named the *projected cost*. For $t = 0$, $\mathcal{L}_0$ is simply the cost of the initial state $s_0$. For $t > 0$, $\mathcal{L}_t$ is the cost of the state projected a duration $t$ in the future.

**Proposition 8** (Gradients of RBP are Gradients of the Projected Cost). *The 'RBP gradients' $\nabla_s^{\mathrm{RBP}}(t)$ and $\nabla_\theta^{\mathrm{RBP}}(t)$ can be expressed as gradients of the projected cost:*

$$\forall t \geq 0, \qquad \nabla_s^{\mathrm{RBP}}(t) = \left.\frac{\partial \mathcal{L}_t}{\partial s_0}\right|_{s_0 = s_*}, \qquad \nabla_\theta^{\mathrm{RBP}}(t) = \left.\frac{\partial \mathcal{L}_t}{\partial \theta_0}\right|_{s_0 = s_*} \tag{36}$$

*where the initial state $s_0$ is the steady state $s_*$.*

*Proof of Proposition 7.* First of all, by Definition 6 (Eq. 32-34) it is straightforward to see that

$$\forall t \geq 0, \qquad \nabla_s^{\mathrm{RBP}}(t) = \left( \frac{\partial F}{\partial s} (x, s_*, \theta)^\top \right)^t \cdot \frac{\partial \ell}{\partial s} (s_*, y) , \tag{37}$$

$$\forall t \geq 1, \qquad \nabla_\theta^{\mathrm{RBP}}(t) = \frac{\partial F}{\partial \theta} (x, s_*, \theta)^\top \cdot \left( \frac{\partial F}{\partial s} (x, s_*, \theta)^\top \right)^{t-1} \cdot \frac{\partial \ell}{\partial s} (s_*, y) . \tag{38}$$

Second, recall that the loss $\mathcal{L}^*$ is
$$\mathcal{L}^* = \ell (s_*, y) , \tag{39}$$
where
$$s_* = F (x, s_*, \theta) . \tag{40}$$
By the chain rule of differentiation, the gradient of $\mathcal{L}^*$ (Eq. 39) is
$$\frac{\partial \mathcal{L}^*}{\partial \theta} = \frac{\partial \ell}{\partial s} (s_*, y) \cdot \frac{\partial s_*}{\partial \theta} . \tag{41}$$

In order to compute $\frac{\partial s_*}{\partial \theta}$, we differentiate the steady state condition (Eq. 40) with respect to $\theta$, which yields
$$\frac{\partial s_*}{\partial \theta} = \frac{\partial F}{\partial s} (x, s_*, \theta) \cdot \frac{\partial s_*}{\partial \theta} + \frac{\partial F}{\partial \theta} (x, s_*, \theta) . \tag{42}$$

Rearranging the terms, and using the Taylor expansion $(\mathrm{Id} - A)^{-1} = \sum_{t=0}^\infty A^t$ with $A = \frac{\partial F}{\partial s} (x, s_*, \theta)$, we get

$$\frac{\partial s_*}{\partial \theta} = \left( \mathrm{Id} - \frac{\partial F}{\partial s} (x, s_*, \theta) \right)^{-1} \cdot \frac{\partial F}{\partial \theta} (x, s_*, \theta) \tag{43}$$

$$= \sum_{t=0}^\infty \left( \frac{\partial F}{\partial s} (x, s_*, \theta) \right)^t \cdot \frac{\partial F}{\partial \theta} (x, s_*, \theta) . \tag{44}$$

Therefore

$$\frac{\partial \mathcal{L}^*}{\partial \theta} = \frac{\partial \ell}{\partial s} (s_*, y) \cdot \frac{\partial s_*}{\partial \theta} \tag{45}$$

$$= \sum_{t=0}^\infty \frac{\partial \ell}{\partial s} (s_*, y) \cdot \left( \frac{\partial F}{\partial s} (x, s_*, \theta) \right)^t \cdot \frac{\partial F}{\partial \theta} (x, s_*, \theta) \tag{46}$$

$$= \sum_{t=0}^\infty \nabla_\theta^{\mathrm{RBP}}(t). \tag{47}$$

$\square$

*Proof of Proposition 8.* By the chain rule of differentiation we have
$$\frac{\partial \mathcal{L}_{t+1}}{\partial s_0} = \frac{\partial F}{\partial s} (x, s_0, \theta)^\top \cdot \frac{\partial \mathcal{L}_{t+1}}{\partial s_1} . \tag{48}$$

Evaluation this expression for $s_0 = s_*$ we get
$$\left. \frac{\partial \mathcal{L}_{t+1}}{\partial s_0} \right|_{s_0 = s_*} = \frac{\partial F}{\partial s} (x, s_*, \theta)^\top \cdot \left. \frac{\partial \mathcal{L}_{t+1}}{\partial s_1} \right|_{s_0 = s_*} . \tag{49}$$

Finally note that
$$\left. \frac{\partial \mathcal{L}_{t+1}}{\partial s_1} \right|_{s_0 = s_*} = \left. \frac{\partial \mathcal{L}_{t+1}}{\partial s_1} \right|_{s_1 = s_*} = \left. \frac{\partial \mathcal{L}_t}{\partial s_0} \right|_{s_0 = s_*} \tag{50}$$

Therefore $\left. \frac{\partial \mathcal{L}_t}{\partial s_0} \right|_{s_0 = s_*}$ and $\nabla_s^{\mathrm{RBP}}(t)$ satisfy the same recurrence relation, thus they are equal. Proving the equality of $\left. \frac{\partial \mathcal{L}_t}{\partial \theta_0} \right|_{s_0 = s_*}$ and $\nabla_\theta^{\mathrm{RBP}}(t)$ is analogous. $\square$

## C    ILLUSTRATING THE EQUIVALENCE OF THE FOUR ALGORITHMS ON AN ANALYTICALLY TRACTABLE MODEL

**Model.**    To illustrate the equivalence of the four algorithms (BPTT, RBP, EP and CEP), we study a simple model with scalar variable $s$ and scalar parameter $\theta$:

$$s_0 = 0, \qquad s_{t+1} = \frac{1}{2}\left(s_t + \theta\right), \qquad \mathcal{L}^* = \frac{1}{2}s_*^2, \tag{51}$$

where $s_*$ is the steady state of the dynamics (it is easy to see that the solution is $s_* = \theta$). The dynamics rewrites $s_{t+1} = F\left(s_t, \theta\right)$ with the transition function $F\left(s, \theta\right) = \frac{1}{2}\left(s + \theta\right)$, and the loss rewrites $\mathcal{L}^* = \ell\left(s_*\right)$ with the cost function $\ell(s) = \frac{1}{2}s^2$. Furthermore, a primitive function of the system [1] is $\Phi(s, \theta) = \frac{1}{4}(s + \theta)^2$. This model has no practical application ; it is only meant for pedagogical purpose.

**Backpropagation Through Time (BPTT).**    With BPTT, an important point is that we approximate the steady state $s_*$ by the state after $T$ time steps $s_T$, and we approximate $\mathcal{L}^*$ (the loss at the steady state) by the loss after $T$ time steps $\mathcal{L} = \ell\left(s_T\right)$.

In order to compute (i.e. 'backpropagate') the gradients of BPTT, Proposition 5 tells us that we need to compute $\frac{\partial \ell}{\partial s}\left(s_T\right) = s_T$, $\frac{\partial F}{\partial s}\left(s_t, \theta\right) = \frac{1}{2}$ and $\frac{\partial F}{\partial \theta}\left(s_t, \theta\right) = \frac{1}{2}$. We get

$$\forall t = 0, 1, \ldots, T - 1, \qquad \nabla_s^{\mathrm{BPTT}}(t) = \frac{s_T}{2^t}, \qquad \nabla_\theta^{\mathrm{BPTT}}(t) = \frac{s_T}{2^{t+1}}. \tag{52}$$

**Recurrent Backpropagation (RBP).**    Similarly, to compute the gradients of RBP, Definition 6 tells us that we need to compute $\frac{\partial \ell}{\partial s}\left(s_*\right) = s_*$, $\frac{\partial F}{\partial s}\left(s_*, \theta\right) = \frac{1}{2}$ and $\frac{\partial F}{\partial \theta}\left(s_*, \theta\right) = \frac{1}{2}$. We have

$$\forall t \geq 0, \qquad \nabla_s^{\mathrm{RBP}}(t) = \frac{s_*}{2^t}, \qquad \nabla_\theta^{\mathrm{RBP}}(t) = \frac{s_*}{2^{t+1}}. \tag{53}$$

The state after $T$ time steps in BPTT converges to the steady state $s_*$ as $T \to \infty$, therefore the gradients of BPTT converge to the gradients of RBP. Also notice that the steady state of the dynamics is $s_* = \theta$.

**Equilibrium Propagation (EP).**    Following the equations governing the second phase of EP (Fig. 1), we have:

$$s_0^\beta = \theta, \qquad s_{t+1}^\beta = \left(\frac{1}{2} - \beta\right) s_t^\beta + \frac{1}{2}\theta. \tag{54}$$

This linear dynamical system can be solved analytically:

$$\forall t \geq 0, \qquad s_t^\beta = \frac{\theta}{1 + 2\beta}\left(1 + 2\beta\left(\frac{1}{2} - \beta\right)^t\right). \tag{55}$$

Notice that $s_t^\beta \to \theta$ as $\beta \to 0$ ; for small values of the hyperparameter $\beta$, the trajectory in the second phase is close to the steady state $s_* = \theta$.

Using Eq. 19, it follows that the normalized updates of EP are

$$\forall t \geq 0, \qquad \Delta_s^{\mathrm{EP}}(\beta, t) = -\frac{\theta}{2^t}\left(1 - 2\beta\right)^t, \qquad \Delta_\theta^{\mathrm{EP}}(\beta, t) = -\frac{\theta}{2^{t+1}}\left(1 - 2\beta\right)^t. \tag{56}$$

Notice again that the normalized updates of EP converge to the gradients of RBP as $\beta \to 0$.

**Continual Equilibrium Propagation (C-EP).**    The system of equations governing the system is:

$$\begin{cases} s_0^{\beta,\eta} = s_*, \\ \theta_0^{\beta,\eta} = \theta, \end{cases} \qquad \forall t \geq 0 : \begin{cases} s_{t+1}^{\beta,\eta} = \left(\frac{1}{2} - \beta\right) s_t^{\beta,\eta} + \frac{1}{2}\theta_t^{\beta,\eta}, \\ \theta_{t+1}^{\beta,\eta} = \theta_t^{\beta,\eta} + \frac{\eta}{2\beta}\left(s_{t+1}^{\beta,\eta} - s_t^{\beta,\eta}\right). \end{cases} \tag{57}$$

---

[1]The primitive function $\Phi$ is determined up to a constant.

First, rearranging the terms in the second equation, we get

$$\frac{1}{\eta}\left(\theta_{t+1}^{\beta,\eta} - \theta_t^{\beta,\eta}\right) = \frac{1}{2\beta}\left(s_{t+1}^{\beta,\eta} - s_t^{\beta,\eta}\right).$$

(58)

It follows that

$$\Delta_\theta^{C-EP}(\beta,\eta,t) = \frac{1}{2}\Delta_s^{C-EP}(\beta,\eta,t).$$

(59)

Therefore, all we need to do is to compute $\Delta_s^{C-EP}(\beta,\eta,t)$. Second, by iterating the second equation over all indices from $t = 0$ to $t - 1$ we get

$$\theta_t^{\beta,\eta} = \theta + \frac{\eta}{2\beta}\left(s_t^{\beta,\eta} - s_*\right).$$

(60)

Using $s_* = \theta$ and plugging this into the first equation we get

$$s_{t+1}^{\beta,\eta} = \left(\frac{1}{2} - \beta + \frac{\eta}{4\beta}\right)s_t^{\beta,\eta} + \left(\frac{1}{2} - \frac{\eta}{4\beta}\right)\theta.$$

(61)

Solving this linear dynamical system, and using the initial condition $s_0^{\beta,\eta} = \theta$ we get

$$s_t^{\beta,\eta} = \frac{\theta}{1 - \frac{\eta}{2\beta} + 2\beta}\left[1 - \frac{\eta}{2\beta} + 2\beta\left(\frac{1}{2}\right)^t\left(1 - 2\beta + \frac{\eta}{2\beta}\right)^t\right]$$

(62)

Finally:

$$\Delta_s^{C-EP}(\beta,\eta,t) = -\frac{\theta}{2^t}\left(1 - 2\beta + \frac{\eta}{2\beta}\right)^t$$

(63)

## D  COMPLEMENT ON GRADIENT-DESCENDING DYNAMICS (GDD)

Step-by-step equivalence of the dynamics of EP and gradient computation in BPTT was shown in Ernoult et al. (2019) and was refered to as the Gradient-Descending Updates (GDU) property. In this appendix, we first explain the connection between the GDD property of this work and the GDU property of Ernoult et al. (2019). Then we prove another version of the GDD property (Theorem 9 below), more general than Theorem 1.

### D.1  GRADIENT-DESCENDING UPDATES (GDU) OF ERNOULT ET AL. (2019)

The GDU property of Ernoult et al. (2019) states that the (normalized) updates of EP are equal to the gradients of BPTT. Similarly, the Gradient-Descending Dynamics (GDD) property of this work states that the normalized updates of C-EP are equal to the gradients of BPTT. The difference between the GDU property and the GDD property is that the term 'update' has slightly different meanings in the contexts of EP and C-EP. In C-EP, the 'updates' are the *effective* updates by which the neuron and synapses are being dynamically updated throughout the second phase. In contrast in EP, the 'updates' are effectively performed at the end of the second phase.

### D.2  A GENERALISATION OF THE GDD PROPERTY

The *Gradient Descending Dynamics* property (GDD, Theorem 1) states that, when the system dynamics derive from a primitive function, i.e. when the transition function $F$ is of the form $F = \frac{\partial \Phi}{\partial s}$, then the normalized updates of C-EP match the gradients provided by BPTT. Remarkably, even in the case of the C-VF dynamics that do not derive from a primitive function $\Phi$, Fig. 5 shows that the biologically realistic update rule of C-VF follows well the gradients of BPTT. More illustrations of this property are shown on Fig. 12 and Fig. 13. In this section we give a theoretical justification for this fact by proving a more general result than Theorem 1.

First, recall the dynamics of the C-VF model. In the first phase:

$$s_{t+1} = \sigma \left( W \cdot s_t \right), \tag{64}$$

where $\sigma$ is an activation function and $W$ is a square weight matrix. In the second phase, starting from $s_0^{\beta,\eta} = s_*$ and $W_0^{\beta,\eta} = W$, the dynamics read:

$$\forall t \geq 0 : \quad \begin{cases} s_{t+1}^{\beta,\eta} = \sigma \left( W_t^{\beta,\eta} \cdot s_t^{\beta,\eta} \right) - \beta \dfrac{\partial \ell}{\partial s} \left( s_t^{\beta,\eta} \right), \\ W_{t+1}^{\beta,\eta} = W_t^{\beta,\eta} + \dfrac{\eta}{\beta} s_t^{\beta,\eta^\top} \cdot \left( s_{t+1}^{\beta,\eta} - s_t^{\beta,\eta} \right). \end{cases} \tag{65}$$

Now let us define the transition function $F(s, W) = \sigma(W \cdot s)$, so that the dynamics of the first phase rewrites

$$s_{t+1} = F \left( s_t, W \right). \tag{66}$$

As for the second phase, notice that $\frac{\partial F}{\partial W}(s, W) = \sigma'(W \cdot s) \cdot s$, so that if we ignore the factor $\sigma'(W \cdot s)$, Eq. (65) rewrites

$$\forall t \geq 0 : \quad \begin{cases} s_{t+1}^{\beta,\eta} = F \left( s_t^{\beta,\eta}, W_t^{\beta,\eta} \right) - \beta \dfrac{\partial \ell}{\partial s} \left( s_t^{\beta,\eta} \right), \\ W_{t+1}^{\beta,\eta} = W_t^{\beta,\eta} + \dfrac{\eta}{\beta} \dfrac{\partial F}{\partial W} \left( s_t^{\beta,\eta}, W_t^{\beta,\eta} \right)^\top \cdot \left( s_{t+1}^{\beta,\eta} - s_t^{\beta,\eta} \right). \end{cases} \tag{67}$$

Now, recall the definition of the normalized updates of C-VF, as well as the gradients of the loss $\mathcal{L} = \ell \left( s_T, y \right)$ after $T$ time steps, computed with BPTT:

$$\begin{cases} \Delta_s^{\mathrm{C-VF}}(\beta, \eta, t) = \dfrac{1}{\beta} \left( s_{t+1}^{\beta,\eta} - s_t^{\beta,\eta} \right), \\ \Delta_W^{\mathrm{C-VF}}(\beta, \eta, t) = \dfrac{1}{\eta} \left( W_{t+1}^{\beta,\eta} - W_t^{\beta,\eta} \right), \end{cases} \qquad \begin{cases} \nabla_s^{\mathrm{BPTT}}(t) = \dfrac{\partial \mathcal{L}}{\partial s_{T-t}}, \\ \nabla_W^{\mathrm{BPTT}}(t) = \dfrac{\partial \mathcal{L}}{\partial W_{T-t}}. \end{cases} \tag{68}$$

The loss $\mathcal{L}$ and the gradients $\nabla_s^{\mathrm{BPTT}}(t)$ and $\nabla_\theta^{\mathrm{BPTT}}(t)$ are defined formally in Appendix B.2.

**Theorem 9** (Generalisation of the GDD Property). *Let $s_0, s_1, \ldots, s_T$ be the convergent sequence of states and denote $s_* = s_T$ the steady state. Further assume that there exists some step $K$ where $0 < K \leq T$ such that $s_* = s_T = s_{T-1} = \ldots s_{T-K}$. Finally, assume that the Jacobian of the transition function at the steady state is symmetric, i.e. $\frac{\partial F}{\partial s}(s_*, W) = \frac{\partial F}{\partial s}(s_*, W)^\top$. Then, in the limit $\eta \to 0$ and $\beta \to 0$, the first $K$ normalized updates of C-VF follow the the first $K$ gradients of BPTT, i.e.*

$$\forall t = 0, 1, \ldots, K : \quad \begin{cases} \lim\limits_{\beta \to 0} \lim\limits_{\eta \to 0} \Delta_s^{\mathrm{C-VF}}(\beta, \eta, t) = -\nabla_s^{\mathrm{BPTT}}(t), \\ \lim\limits_{\beta \to 0} \lim\limits_{\eta \to 0} \Delta_W^{\mathrm{C-VF}}(\beta, \eta, t) = -\nabla_W^{\mathrm{BPTT}}(t). \end{cases} \tag{69}$$

A few remarks need to be made:

1. Observe that

$$\frac{\partial F}{\partial s}(s, W) = \sigma'(W \cdot s) \cdot W^\top. \tag{70}$$

   Ignoring the factor $\sigma'(W \cdot s)$, we see that if $W$ is symmetric then the Jacobian of $F$ is also symmetric, in which case the conditions of Theorem 9 are met.

2. Theorem 1 is a special case of Theorem 9. To see why, notice that if the transition function $F$ is of the form $F(s, W) = \frac{\partial \Phi}{\partial s}(s, W)$, then

$$\frac{\partial F}{\partial s}(s, W) = \frac{\partial^2 \Phi}{\partial s^2}(s, W) = \frac{\partial F}{\partial s}(s, W)^\top \tag{71}$$

   In this case the extra assumption in Theorem 9 is automatically satisfied.

### D.3 PROOF OF THEOREM 9

Theorem 9 is a consequence of Proposition 5 (Appendix B.2), which we recall here, and Lemma 10 below.

**Proposition 5** (Gradients of BPTT). *The gradients $\nabla_s^{\mathrm{BPTT}}(t)$ and $\nabla_\theta^{\mathrm{BPTT}}(t)$ satisfy the recurrence relationship*

$$\nabla_s^{\mathrm{BPTT}}(0) = \frac{\partial \ell}{\partial s}(s_T, y), \tag{29}$$

$$\forall t = 1, 2, \ldots, T, \quad \nabla_s^{\mathrm{BPTT}}(t) = \frac{\partial F}{\partial s}(x, s_{T-t}, \theta)^\top \cdot \nabla_s^{\mathrm{BPTT}}(t-1), \tag{30}$$

$$\forall t = 1, 2, \ldots, T, \quad \nabla_\theta^{\mathrm{BPTT}}(t) = \frac{\partial F}{\partial \theta}(x, s_{T-t}, \theta)^\top \cdot \nabla_s^{\mathrm{BPTT}}(t-1). \tag{31}$$

**Lemma 10** (Updates of C-VF). *Define the (normalized) neural and weight updates of C-VF in the limit $\eta \to 0$ and $\beta \to 0$:*

$$\forall t = 0, 1, \ldots, K : \quad \begin{cases} \Delta_s^{\mathrm{C-VF}}(t) = \lim\limits_{\beta \to 0} \lim\limits_{\eta \to 0} \Delta_s^{\mathrm{C-VF}}(\beta, \eta, t), \\ \Delta_\theta^{\mathrm{C-VF}}(t) = \lim\limits_{\beta \to 0} \lim\limits_{\eta \to 0} \Delta_\theta^{\mathrm{C-VF}}(\beta, \eta, t). \end{cases} \tag{72}$$

*They satisfy the recurrence relationship*

$$\Delta_s^{\mathrm{C-VF}}(0) = -\frac{\partial \ell}{\partial s}(s_*, y), \tag{73}$$

$$\forall t \geq 0, \quad \Delta_s^{\mathrm{C-VF}}(t+1) = \frac{\partial F}{\partial s}(x, s_*, \theta) \cdot \Delta_s^{\mathrm{C-VF}}(t), \tag{74}$$

$$\forall t \geq 0, \quad \Delta_\theta^{\mathrm{C-VF}}(t+1) = \frac{\partial F}{\partial \theta}(x, s_*, \theta)^\top \cdot \Delta_s^{\mathrm{C-VF}}(t). \tag{75}$$

The proof of Lemma 10 is similar to the one provided in Ernoult et al. (2019).

# E    MODELS

In this section, we describe the C-EP and C-VF algorithms when implemented on multi-layered models, with tied weights and untied weights respectively. In the fully connected layered architecture model, the neurons are only connected between two consecutive layers (no skip-layer connections and no lateral connections within a layer). We denote neurons of the n-th layer as $s^n$ with $n \in [0, N-1]$, where $N$ is the number of hidden layers. Layers are labelled in a backward fashion: $n = 0$ labels the output layer, $n = 1$ the first hidden layer starting from the output layer, and $n = N - 1$ the last hidden layer (before the input layer). Thus, there are $N$ hidden layers in total. Fig. 7 shows this architecture with $N = 2$. Each model are presented here in a "real-time" and "discrete-time" settings For each model we lay out the equations of the neuron and synapse dynamics, we demonstrate the GDD property and we specify in which part of the main text they are used.

We present in this order:

1. Discrete-Time RNN with symmetric weights trained with EP,

2. Discrete-Time RNN with symmetric weights trained with C-EP,

3. Real-Time RNN with symmetric weights trained with C-EP,

4. Discrete-Time RNN with asymmetric weights trained with C-VF,

5. Real-Time RNN with asymmetric weights trained with C-VF.

Our 'Discrete-Time RNN model' is also commonly called 'vanilla RNN', and it is refered to as the 'prototypical model' in Ernoult et al. (2019). Our 'Real-Time RNN model with symmetric weights' is also commonly called 'continuous Hopfield model' and is refered to as the 'energy-based model' in Ernoult et al. (2019).

**Demonstrating the Gradient Descending Dynamics (GDD) property (Theorem 1) on MNIST.** For this experiment, we consider the 784-512-...-512-10 network architecture, with 784 input neurons, 10 ouput neurons, and 512 neurons per hidden layer. The activation function used is $\sigma(x) = \tanh(x)$. The experiment consists of the following: we take a random MNIST sample (of size $1 \times 784$) and its associated target (of size $1 \times 10$). For a given value of the time-discretization parameter $\epsilon$, we perform the first phase for $T$ steps. Then, we perform on the one hand BPTT over $K$ steps (to compute the gradients $\nabla^{\mathrm{BPTT}}$), on the other hand C-EP (or C-VF) over $K$ steps for given values of $\beta$ and $\eta$ (to compute the normalized updates $\Delta^{\mathrm{C-EP}}$ or $\Delta^{\mathrm{C-VF}}$) and compare the gradients and normalized updates provided by the two algorithms. Precise values of the hyperparameters $\epsilon$, $T$, $K$, $\beta$ and $\eta$ are given in Tab. E.6.

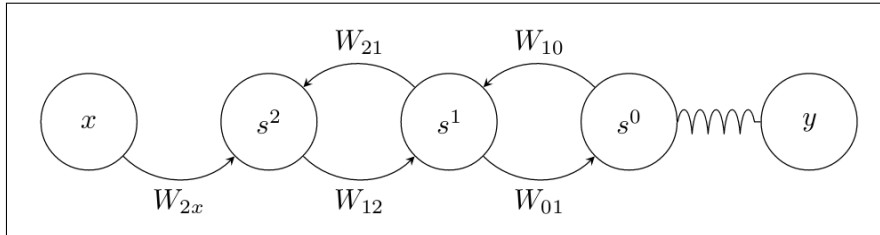

Figure 7: Fully connected layered architecture with $N = 2$ hidden layers.

## E.1    DISCRETE-TIME RNN WITH SYMMETRIC WEIGHTS TRAINED WITH EP

**Context of use.**    This model is used for training experiments in Section 4.2 and Table 4.1.

**Equations with $N = 2$.**    We consider the layered architecture of Fig. 7, where $s^0$ denotes the output layer, and the feedback connections are constrained to be the transpose of the feedforward connections, i.e. $W_{nn-1} = W_{n-1n}^\top$. In the discrete-time setting of EP, the dynamics of the first phase

are defined as:

$$\forall t \in [0, T] : \begin{cases} s_{t+1}^0 = \sigma\left(W_{01} \cdot s_t^1\right), \\ s_{t+1}^1 = \sigma\left(W_{12} \cdot s_t^2 + W_{01}^\top \cdot s_t^0\right), \\ s_{t+1}^2 = \sigma\left(W_{2x} \cdot x + W_{12}^\top \cdot s_t^1\right). \end{cases} \tag{76}$$

In the second phase the dynamics reads:

$$\forall t \in [0, K] : \begin{cases} s_{t+1}^{0,\beta} = \sigma\left(W_{01} \cdot s_t^{1,\beta}\right) + \beta\,\epsilon\,\left(y - s_t^{0,\beta}\right), \\ s_{t+1}^{1,\beta} = \sigma\left(W_{12} \cdot s_t^{2,\beta} + W_{01}^\top \cdot s_t^{0,\beta}\right), \\ s_{t+1}^{2,\beta} = \sigma\left(W_{2x} \cdot x + W_{12}^\top \cdot s_t^{1,\beta}\right). \end{cases} \tag{77}$$

As usual, $y$ denotes the target. Consider the function:

$$\Phi\left(x, s^2, s^1, s^0\right) = (s^0)^\top \cdot W_{01} \cdot s^1 + (s^1)^\top \cdot W_{12} \cdot s^2 + (s^2)^\top \cdot W_{2x} \cdot x. \tag{78}$$

We can compute, for example:

$$\frac{\partial \Phi}{\partial s^1} = W_{12} \cdot s^2 + W_{01}^\top \cdot s^0. \tag{79}$$

Comparing Eq. (76) and Eq. (79), and ignoring the activation function $\sigma$, we can see that

$$s_t^1 \approx \frac{\partial \Phi}{\partial s^1}\left(x, s_{t-1}^2, s_{t-1}^1, s_{t-1}^0\right). \tag{80}$$

And similarly for the layers $s^0$ and $s^2$.

According to the definition of $\Delta_\theta^{\mathrm{C-EP}}$ in Eq. (19), for every layer and every $t \in [0, K]$:

$$\begin{cases} \Delta_{W_{2x}}^{\mathrm{EP}}(\beta, t) = \frac{1}{\beta}\left(s_{t+1}^{2,\beta} \cdot x^\top - s_t^{2,\beta} \cdot x^\top\right), \\ \Delta_{W_{12}}^{\mathrm{EP}}(\beta, t) = \frac{1}{\beta}\left(s_{t+1}^{1,\beta} \cdot s_{t+1}^{2,\beta^\top} - s_t^{1,\beta} \cdot s_t^{2,\beta^\top}\right), \\ \Delta_{W_{01}}^{\mathrm{EP}}(\beta, t) = \frac{1}{\beta}\left(s_{t+1}^{0,\beta} \cdot s_{t+1}^{1,\beta^\top} - s_t^{0,\beta} \cdot s_t^{1,\beta^\top}\right). \end{cases} \tag{81}$$

**Simplifying the equations with $N = 2$.** To go from our multi-layered architecture to the more general model presented in section 4.1. we define the state $s$ of the network as the concatenation of all the layers' states, i.e. $s = (s^2, s^1, s^0)^\top$ and we define the weight matrices $W$ and $W_x$ as:

$$W = \begin{pmatrix} 0 & W_{12}^\top & 0 \\ W_{12} & 0 & W_{01}^\top \\ 0 & W_{01} & 0 \end{pmatrix}, \qquad W_x = \begin{pmatrix} W_{2x} \\ 0 \\ 0 \end{pmatrix}. \tag{82}$$

Note that Eq. (76) and Eq. (78) can be vectorized into:

$$s_{t+1} = \sigma(W \cdot s_t + W_x \cdot x), \tag{83}$$

$$\Phi = \frac{1}{2} s^T \cdot W \cdot s + \frac{1}{2} s^T \cdot W_x \cdot x. \tag{84}$$

**Generalizing the equations for any $N$.** For a general architecture with a given $N$, the dynamics of the first phase are defined as:

$$\forall t \in [0, T] : \begin{cases} s_{t+1}^0 = \sigma\left(W_{01} \cdot s_t^1\right) \\ s_{t+1}^n = \sigma\left(W_{nn+1} \cdot s_t^{n+1} + W_{n-1n}^\top \cdot s_t^{n-1}\right) \qquad \forall n \in [1, N-1] \\ s_{t+1}^N = \sigma\left(W_{N,x} \cdot x + W_{N-1N}^\top \cdot s_t^{N-1}\right), \end{cases} \tag{85}$$

and those of the second phase as:

$$\forall t \in [0, K] : \begin{cases} s_{t+1}^{0,\beta} = \sigma\left(W_{01} \cdot s_t^{1,\beta}\right) + \beta(y - s^{0,\beta}) \\ s_{t+1}^{n,\beta} = \sigma\left(W_{nn+1} \cdot s_t^{n+1,\beta} + W_{n-1n}^\top \cdot s_t^{n-1,\beta}\right) \qquad \forall n \in [1, N-1] \\ s_{t+1}^{N,\beta} = \sigma\left(W_{N,x} \cdot x + W_{N-1N}^\top \cdot s_t^{N-1,\beta}\right), \end{cases} \tag{86}$$

where $y$ denotes the target. Defining:

$$\Phi(x, s^N, \ldots, s^0) = \sum_{n=0}^{N-1} s^{n^\top} \cdot W_{nn+1} \cdot s^{n+1} + s^N \cdot W_{N,x} \cdot x, \tag{87}$$

ignoring the activation function $\sigma$, Eq. (85) rewrites:

$$s_{t+1}^n \approx \frac{\partial \Phi}{\partial s^n}(x, s^N, \ldots, s^0) \qquad \forall n \in [1, N-1] \tag{88}$$

According to the definition of $\Delta_\theta^{\text{C-EP}}$ in Eq. (19), for every layer $W_{nn+1}$ and every $t \in [0, K]$:

$$\begin{cases} \Delta_{W_{N,x}}^{\text{EP}}(\beta, t) = \frac{1}{\beta}\left(s_{t+1}^{N,\beta} \cdot x^\top - s_t^{N,\beta} \cdot x^\top\right), \\ \Delta_{W_{nn+1}}^{\text{EP}}(\beta, t) = \frac{1}{\beta}\left(s_{t+1}^{n,\beta} \cdot s_{t+1}^{n+1,\beta^\top} - s_t^{n,\beta} \cdot s_t^{n+1,\beta^\top}\right) \qquad \forall n \in [0, N-1] \end{cases} \tag{89}$$

Defining $s = (s^N, s^{N-1}, \ldots, s^0)^\top$ and:

$$W = \begin{pmatrix} 0 & W_{N-1N}^\top & 0 & 0 & 0 \\ W_{N-1N} & 0 & W_{N-2N-1}^\top & 0 & 0 \\ 0 & W_{N-2N-1} & 0 & \ddots & 0 \\ 0 & 0 & \ddots & 0 & W_{01}^\top \\ 0 & 0 & 0 & W_{01} & 0 \end{pmatrix}, \qquad W_x = \begin{pmatrix} W_{N,x} \\ 0 \\ \vdots \\ 0 \end{pmatrix}, \tag{90}$$

Eq. (85) and Eq. (87) can also be vectorized into:

$$s_{t+1} = \sigma(W \cdot s_t + W_x \cdot x) \tag{91}$$

$$\Phi(x, s, W, W_x) = \frac{1}{2}s^T \cdot W \cdot s + \frac{1}{2}s^T \cdot W_x \cdot x. \tag{92}$$

Thereafter we introduce the other models in this general case.

### E.2 DISCRETE-TIME RNN WITH SYMMETRIC WEIGHTS TRAINED WITH C-EP

**Context of use.** This model is used for training experiments in Section 4.2 and Table 4.1.

**Equations.** Recall that we consider the layered architecture of Fig. 7, where $s^0$ denotes the output layer. Just like in the discrete-time setting of EP, the dynamics of the first phase are defined as:

$$\forall t \in [0, T] : \begin{cases} s_{t+1}^0 = \sigma\left(W_{01} \cdot s_t^1\right) \\ s_{t+1}^n = \sigma\left(W_{nn+1} \cdot s_t^{n+1} + W_{n-1n}^\top \cdot s_t^{n-1}\right) \qquad \forall n \in [1, N-1] \\ s_{t+1}^N = \sigma\left(W_{N,x} \cdot x + W_{N-1N}^\top \cdot s_t^{N-1}\right) \end{cases} \tag{93}$$

Again, as in EP, the feedback connections are constrained to be the transpose of the feedforward connections, i.e. $W_{nn-1} = W_{n-1n}^\top$. In the second phase the dynamics reads:

$$\forall t \in [0, K] : \begin{cases} s_{t+1}^{0,\beta,\eta} & = \sigma(W_{01} \cdot s_t^{1,\beta,\eta}) + \beta \left(y - s_t^{0,\beta,\eta}\right) \\ s_{t+1}^{n,\beta,\eta} & = \sigma(W_{nn+1} \cdot s_t^{n+1,\beta,\eta} + W_{n-1n}^\top \cdot s_t^{n-1,\beta,\eta}) \qquad \forall n \in [1, N-1], \\ s_{t+1}^{N,\beta,\eta} & = \sigma\left(W_{N,x} \cdot x + W_{N-1N}^\top \cdot s_t^{N-1,\beta,\eta}\right) \\ \theta_{t+1}^{\beta,\eta} & = \theta_t^{\beta,\eta} + \eta\Delta_\theta^{C-EP}(\beta, \eta, t) \qquad \forall \theta \in \{W_{nn+1}\} \end{cases}$$

(94)

As usual, $y$ denotes the target. Since Eq. (93) and Eq. (85) are the same, the equations describing the C-EP model can also be written in a vectorized block-wise fashion, as in Eq. (91) and Eq. (92). We can consequently define the C-EP model in Section 4.1 per Eq. (10).

According to the definitions of Eq. (6) and Eq. (7), for every layer $W_{nn+1}$ and every $t \in [0, K]$:

$$\begin{cases} \Delta_{W_{N,x}}^{C-EP}(\beta, t) = \frac{1}{\beta}\left(s_{t+1}^{N,\beta,\eta} \cdot x^\top - s_t^{N,\beta,\eta} \cdot x^\top\right), \\ \Delta_{W_{nn+1}}^{C-EP}(\beta, t) = \frac{1}{\beta}\left(s_{t+1}^{n,\beta,\eta} \cdot s_{t+1}^{n+1,\beta,\eta^\top} - s_t^{n,\beta,\eta} \cdot s_t^{n+1,\beta,\eta^\top}\right) \qquad \forall n \in [0, N-1] \end{cases}$$

(95)

### E.3 REAL-TIME RNN WITH SYMMETRIC WEIGHTS TRAINED WITH C-EP

**Context of use.** This model has not been used in this work. We only introduce it for completeness with respect to Ernoult et al. (2019).

**Equations.** For this model, the primitive function is defined as:

$$\Phi\left(s^0, s^1, \ldots, s^{N-1}\right) = \frac{1}{2}(1-\epsilon)\left(\sum_{n=0}^N ||s^n||^2\right) + \epsilon \sum_{n=0}^{N-1} \sigma(s^n) \cdot W_{nn+1} \cdot \sigma(s^{n+1}) + \sigma(s^N) \cdot W_{N,x} \cdot \sigma(x)$$

(96)

so that the equations of motion read:

$$\forall t \in [0, T] : \begin{cases} s_{t+1}^0 & = (1 - \epsilon)s_t^0 + \epsilon W_{01} \cdot \sigma(s_t^1) \\ s_{t+1}^n & = (1 - \epsilon)s_t^n + \epsilon(W_{nn+1} \cdot \sigma\left(s_t^{n+1}\right) + W_{n-1n}^\top \cdot \sigma(s_t^{n-1})) \qquad \forall n \in [1, N-1] \\ s_{t+1}^N & = (1 - \epsilon)s_t^N + \epsilon(W_{N,x} \cdot \sigma(x) + W_{N-1N}^\top \cdot \sigma(s_t^{N-1})) \end{cases}$$

In the second phase:

$$\forall t \in [0, K] : \begin{cases} s_{t+1}^{0,\beta,\eta} & = (1 - \epsilon)s_t^{0,\beta,\eta} + \epsilon W_{01} \cdot \sigma(s_t^{1,\beta,\eta}) + \beta\epsilon(y - s^{0,\beta,\eta}(t)) \\ s_{t+1}^{n,\beta,\eta} & = (1 - \epsilon)s_t^{n,\beta,\eta} + \epsilon(W_{nn+1} \cdot \sigma(s_t^{n+1,\beta,\eta}) + W_{n-1n}^\top \cdot \sigma(s_t^{n-1,\beta,\eta})) \qquad \forall n \in [1, N-1], \\ s_{t+1}^{N,\beta,\eta} & = (1 - \epsilon)s_t^{N,\beta,\eta} + \epsilon(W_{N,x} \cdot \sigma(x) + W_{N-1N}^\top \cdot \sigma(s_t^{N-1,\beta,\eta})) \\ \theta_{t+1}^{\beta,\eta} & = \theta_t^{\beta,\eta} + \eta\Delta_\theta^{C-EP}(\beta, \eta, t) \qquad \forall \theta \in \{W_{nn+1}\} \end{cases}$$

(97)

where $\epsilon$ is a time-discretization parameter and $y$ denotes the target.

According the definition of the C-EP dynamics (Eq. (6)), the definition of $\Delta_\theta^{C-EP}$ (Eq. (7)) and the explicit form of $\Phi$ (Eq. 96), for all time step $t \in [0, K]$, we have:

$$\begin{cases} \Delta_{W_{N,x}}^{C-EP}(\beta, \eta, t) & = \frac{1}{\beta}\left(\sigma\left(s_{t+1}^{N,\beta,\eta}\right) \cdot \sigma(x)^\top - \sigma\left(s_t^{N,\beta,\eta}\right) \cdot \sigma(x)^\top\right) \\ \Delta_{W_{nn+1}}^{C-EP}(\beta, \eta, t) & = \frac{1}{\beta}\left(\sigma\left(s_{t+1}^{n,\beta,\eta}\right) \cdot \sigma\left(s_{t+1}^{n+1,\beta,\eta}\right)^\top - \sigma\left(s_t^{n,\beta,\eta}\right) \cdot \sigma\left(s_t^{n+1,\beta,\eta}\right)^\top\right) \qquad \forall n \in [0, N-1] \end{cases}$$

### E.4 DISCRETE-TIME RNN WITH ASYMMETRIC WEIGHTS TRAINED WITH C-VF

**Context of use.** This model is used for training experiments in Section 4.3 and Table 4.1.

**Equations.** Recall that we consider the layered architecture of Fig. 7, where $s^0$ denotes the output layer. The dynamics of the first phase in C-VF are defined as:

$$\forall t \in [0, T] : \begin{cases} s_{t+1}^0 & = \sigma(W_{01} \cdot s_t^1) \\ s_{t+1}^n & = \sigma(W_{nn+1} \cdot s_t^{n+1} + W_{nn-1} \cdot s_t^{n-1}) \qquad \forall n \in [1, N-1] \\ s_{t+1}^N & = \sigma\left(W_{N,x} \cdot x + W_{NN-1} \cdot s_t^{N-1}\right) \end{cases} \qquad (98)$$

Here, note the difference with EP and C-EP: the feedforward and feedback connections are unconstrained. In the second phase of C-VF:

$$\forall t \in [0, K] : \begin{cases} s_{t+1}^{0,\beta,\eta} & = \sigma(W_{01} \cdot s_t^{1,\beta,\eta}) + \beta\epsilon(y - s_t^{0,\beta,\eta}) \\ s_{t+1}^{n,\beta,\eta} & = \sigma(W_{nn+1} \cdot s_t^{n+1,\beta,\eta} + W_{nn-1} \cdot s_t^{n-1,\beta,\eta}) \qquad \forall n \in [1, N-1], \\ s_{t+1}^{N,\beta,\eta} & = \sigma\left(W_{N,x} \cdot x + W_{NN-1} \cdot s_t^{N-1,\beta,\eta}\right) \\ \theta_{t+1}^{\beta,\eta} & = \theta_t^{\beta,\eta} + \eta\Delta_\theta^{\mathrm{C-EP}}(\beta,\eta,t) \qquad \forall \theta \in \{W_{nn+1}, W_{n+1n}\} \end{cases}$$

$$(99)$$

As usual $y$ denotes the target. Note that Eq. (98) can also be in a vectorized block-wise fashion as Eq. (91) with $s = (s^0, s^1, \ldots, s^{N-1})^\top$ and provided that we define $W$ and $W_x$ as:

$$W = \begin{pmatrix} 0 & W_{NN-1} & 0 & 0 & 0 \\ W_{N-1N} & 0 & W_{N-1N-2} & 0 & 0 \\ 0 & W_{N-2N-1} & 0 & \ddots & 0 \\ 0 & 0 & \ddots & 0 & W_{10} \\ 0 & 0 & 0 & W_{01} & 0 \end{pmatrix}, \qquad W_x = \begin{pmatrix} W_{N,x} \\ 0 \\ \vdots \\ 0 \end{pmatrix}, \qquad (100)$$

For all layers $W_{nn+1}$ and $W_{n+1n}$, and every $t \in [0, K]$, we define:

$$\begin{cases} \Delta_{W_{N,x}}^{\mathrm{C-VF}}(\beta,\eta,t) & = \frac{1}{\beta}(s_{t+1}^{N,\beta,\eta} - s_t^{N,\beta,\eta}) \cdot x \\ \Delta_{W_{nn+1}}^{\mathrm{C-VF}}(\beta,\eta,t) & = \frac{1}{\beta}(s_{t+1}^{n,\beta,\eta} - s_t^{n,\beta,\eta}) \cdot s_t^{n+1,\beta,\eta^\top} \\ \Delta_{W_{n+1n}}^{\mathrm{C-VF}}(\beta,\eta,t) & = \frac{1}{\beta}(s_{t+1}^{n+1,\beta,\eta} - s_t^{n+1,\beta,\eta}) \cdot s_t^{n,\beta,\eta^\top} \end{cases}$$

### E.5 REAL-TIME RNN WITH ASYMMETRIC WEIGHTS TRAINED WITH C-VF

**Context of use.** This model is used to generate Fig. 5 - see Table E.6 for precise hyperparameters.

**Equations.** For this model, the dynamics of the first phase are defined as:

$$\forall t \in [0, T] : \begin{cases} s_{t+1}^0 = (1-\epsilon)s_t^0 + \epsilon W_{01} \cdot \sigma\left(s_t^1\right) \\ s_{t+1}^n = (1-\epsilon)s_t^n + \epsilon\left(W_{nn+1} \cdot \sigma\left(s_t^{n+1}\right) + W_{nn-1} \cdot \sigma\left(s_t^{n-1}\right)\right) \qquad \forall n \in [1, N-1] \\ s_{t+1}^N = (1-\epsilon)s_t^N + \epsilon\left(W_{N,x} \cdot \sigma(x) + W_{NN-1} \cdot \sigma\left(s_t^{N-1}\right)\right) \end{cases}$$

where $\epsilon$ is the time-discretization parameter. Again, as in the discre-time version of C-VF, the feedforward and feedback connections $W_{nn-1}$ and $W_{n-1n}$ are unconstrained. In the second phase, the dynamics reads:

$$\forall t \in [0, K] : \begin{cases} s_{t+1}^{0,\beta,\eta} = (1-\epsilon)s_t^{0,\beta,\eta} + \epsilon W_{01} \cdot \sigma\left(s_t^{1,\beta,\eta}\right) + \beta\epsilon\left(y - s_t^{0,\beta,\eta}\right) \\ s_{t+1}^{n,\beta,\eta} = (1-\epsilon)s_t^{n,\beta,\eta} + \epsilon\left(W_{nn+1} \cdot \sigma\left(s_t^{n+1,\beta,\eta}\right) + W_{nn-1} \cdot \sigma\left(s_t^{n-1,\beta,\eta}\right)\right) \qquad \forall n \in [1, N-1] \\ s_{t+1}^{N,\beta,\eta} = (1-\epsilon)s_t^{N,\beta,\eta} + \epsilon\left(W_{N,x} \cdot \sigma(x) + W_{NN-1} \cdot \sigma\left(s_t^{N-1,\beta,\eta}\right)\right) \\ \theta_{t+1}^{\beta,\eta} = \theta_t^{\beta,\eta} + \eta\,\Delta_\theta^{\mathrm{C-VF}}(\beta,\eta,t) \qquad \forall \theta \in \{W_{nn+1}, W_{n+1n}\} \end{cases}$$

$$(101)$$

where $y$ denotes the target, as usual. For every feedforward connection matrix $W_{nn+1}$ and every feedback connection matrix $W_{n+1n}$, and for every time step $t \in [0, K]$ in the second phase, we define

$$\begin{cases} \Delta_{W_{N,x}}^{\mathrm{C-VF}}(\beta,\eta,t) = \frac{1}{\beta}(s_{t+1}^{N,\beta,\eta} - s_t^{N,\beta,\eta}) \cdot \sigma(x) \\ \Delta_{W_{nn+1}}^{\mathrm{C-VF}}(\beta,\eta,t) = \frac{1}{\beta}\left(s_{t+1}^{n,\beta,\eta} - s_t^{n,\beta,\eta}\right) \cdot \sigma\left(s_t^{n+1,\beta,\eta}\right)^\top \\ \Delta_{W_{n+1n}}^{\mathrm{C-VF}}(\beta,\eta,t) = \frac{1}{\beta}\left(s_{t+1}^{n+1,\beta,\eta} - s_t^{n+1,\beta,\eta}\right) \cdot \sigma\left(s_t^{n,\beta,\eta}\right)^\top \end{cases}$$

### E.6 FIGURES FOR THE GDD EXPERIMENTS

In the following figures, we show the effect of using continual updates with a finite learning rate in terms of the $\Delta^{\mathrm{C-EP}}$ and $-\nabla^{\mathrm{BPTT}}$ processes on different models introduced above. These figures have been realized either in the discrete-time or continuous-time setting with the fully connected layered architecture with one hidden layer on MNIST. Dashed an continuous lines respectively represent the normalized updates $\Delta$ and the gradients $\nabla^{\mathrm{BPTT}}$. Each randomly selected synapse or neuron correspond to one color. We add an s or $\theta$ index to specify whether we analyse neuron or synapse updates and gradients. Each C-VF simulation has been realized with an angle between forward and backward weights of 0 degrees (i.e. $\Psi(\theta_{\mathrm{f}}, \theta_{\mathrm{b}}) = 0°$). For each figure, left panels demonstrate the GDD property with C-EP with $\eta = 0$ and the right panels show that, upon using $\eta > 0$, dashed and continuous lines start to split appart.

Table 1: Table of hyperparameters used to demonstrate Theorem 1.

|  | Figure | Angle $\Psi$ (°) | Activation | T | K | $\beta$ | $\epsilon$ | Learning rates |
|---|---|---|---|---|---|---|---|---|
| C-EP | 5 | 0 | tanh | 800 | 80 | 0.01 | 0.08 | $0 - 0$ |
| C-VF | 5 | 45 | tanh | 800 | 80 | 0.01 | 0.08 | $0 - 0$ |
| C-EP | 5 | 0 | tanh | 800 | 80 | 0.01 | 0.08 | $1.510^{-5} - 1.510^{-5}$ |
| C-VF | 5 | 45 | tanh | 800 | 80 | 0.01 | 0.08 | $1.510^{-5} - 1.510^{-5}$ |
| C-VF | 14-15 | 0 | tanh | 800 | 80 | 0.005 | 0.08 | $0 - 0$ |
| C-VF | 14-15 | 0 | tanh | 800 | 80 | 0.005 | 0.08 | $2.10^{-5} - 2.10^{-5}$ |
| C-VF | 12-13 | 0 | tanh | 150 | 10 | 0.01 | $-$ | $0 - 0$ |
| C-VF | 12-13 | 0 | tanh | 150 | 10 | 0.01 | $-$ | $2.10^{-5} - 2.10^{-5}$ |
| C-EP | 10-11 | 0 | tanh | 800 | 80 | 0.05 | 0.08 | $0 - 0$ |
| C-EP | 10-11 | 0 | tanh | 800 | 80 | 0.05 | 0.08 | $2.10^{-5} - 2.10^{-5}$ |
| C-EP | 8-9 | 0 | tanh | 150 | 10 | 0.01 | $-$ | $0 - 0$ |
| C-EP | 8-9 | 0 | tanh | 150 | 10 | 0.01 | $-$ | $2.10^{-5} - 2.10^{-5}$ |

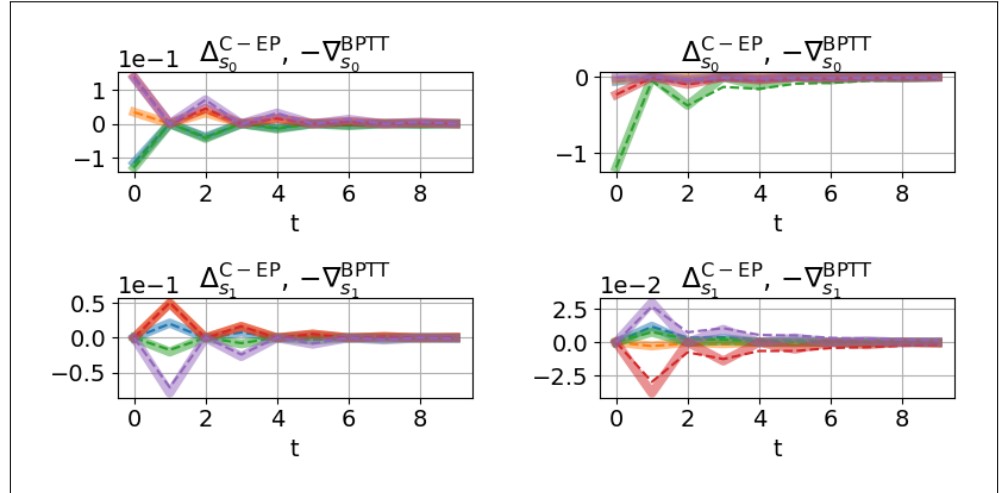

Figure 8: Discrete-Time RNN with symmetric weights. **Left**: $\Delta_s^{\mathrm{C-EP}}(t)$ normalized updates ($\eta = 0$) and $-\nabla_s^{\mathrm{BPTT}}(t)$ gradients. **Right**: $\Delta_s^{\mathrm{C-EP}}(t)$ normalized updates ($\eta > 0$) and $-\nabla_s^{\mathrm{BPTT}}(t)$ gradients.

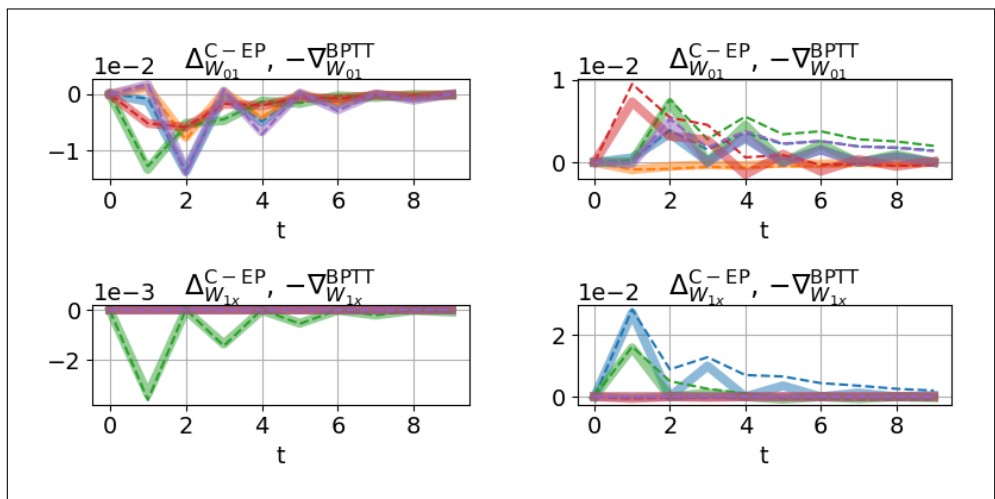

Figure 9: Discrete-Time RNN with symmetric weights. **Left**: $\Delta_\theta^{\mathrm{C-EP}}(t)$ normalized updates ($\eta = 0$) and $-\nabla_\theta^{\mathrm{BPTT}}(t)$ gradients. **Right**: $\Delta_\theta^{\mathrm{C-EP}}(t)$ normalized updates ($\eta > 0$) and $-\nabla_\theta^{\mathrm{BPTT}}(t)$ gradients.

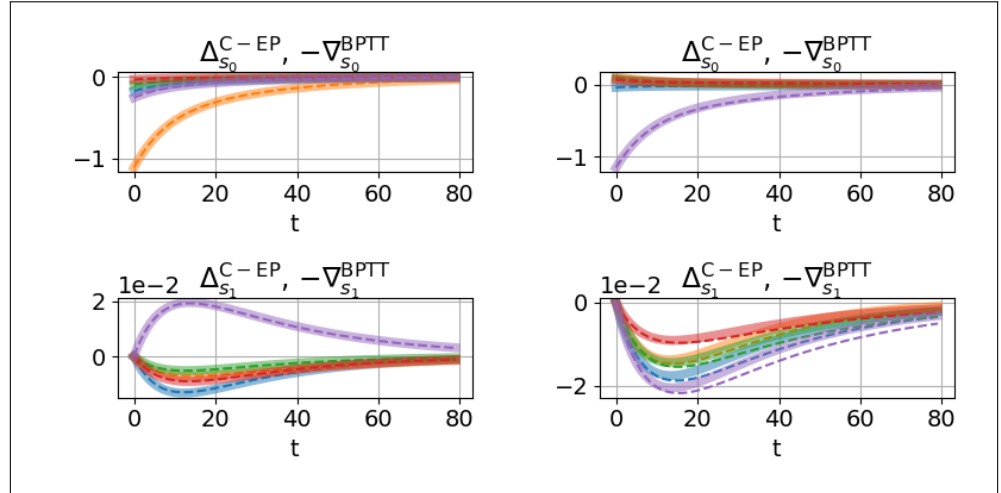

Figure 10: Real-Time RNN with symmetric weights. **Left**: $\Delta_s^{\text{C}-\text{EP}}(t)$ normalized updates ($\eta = 0$) and $-\nabla_s^{\text{BPTT}}(t)$ gradients. **Right**: $\Delta_s^{\text{C}-\text{EP}}(t)$ normalized updates ($\eta > 0$) and $-\nabla_s^{\text{BPTT}}(t)$ gradients.

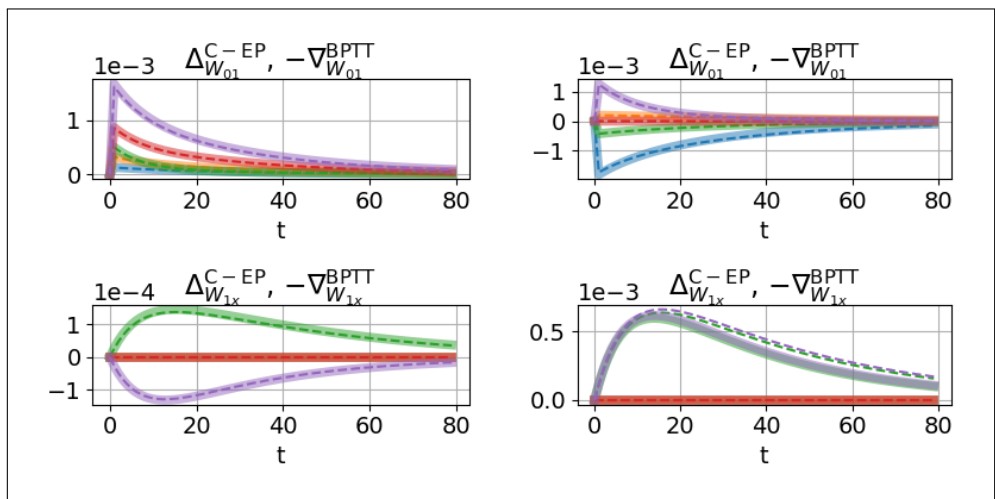

Figure 11: Real-Time RNN with symmetric weights. **Left**$\Delta_\theta^{\text{C}-\text{EP}}(t)$ normalized updates ($\eta = 0$) and $-\nabla_\theta^{\text{BPTT}}(t)$ gradients. **Right**: $\Delta_\theta^{\text{C}-\text{EP}}(t)$ normalized updates ($\eta > 0$) and $-\nabla_\theta^{\text{BPTT}}(t)$ gradients.

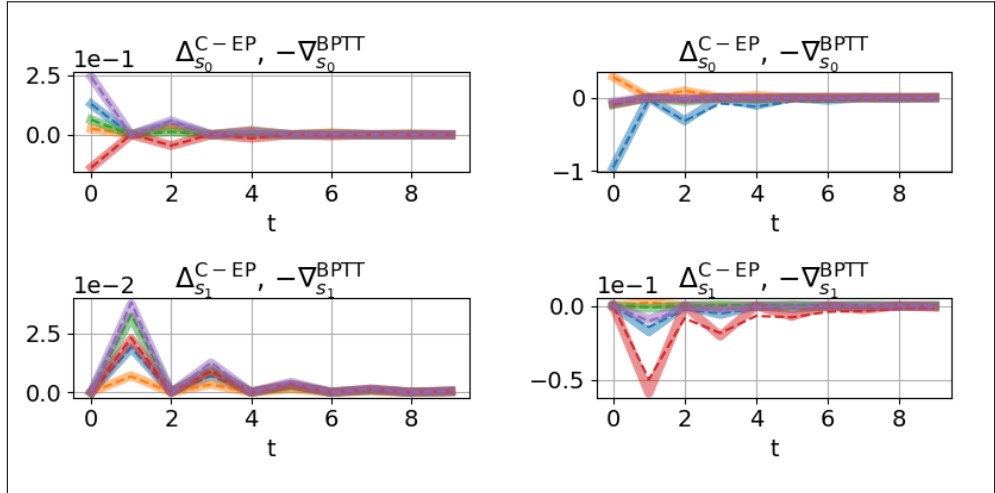

Figure 12: Discrete-Time RNN with asymmetric weights. **Left**: $\Delta_s^{\mathrm{C-VF}}(t)$ normalized updates ($\eta = 0$) and $-\nabla_s^{\mathrm{BPTT}}(t)$ gradients. **Right**: $\Delta_s^{\mathrm{C-VF}}(t)$ normalized updates ($\eta > 0$) and $-\nabla_s^{\mathrm{BPTT}}(t)$ gradients.

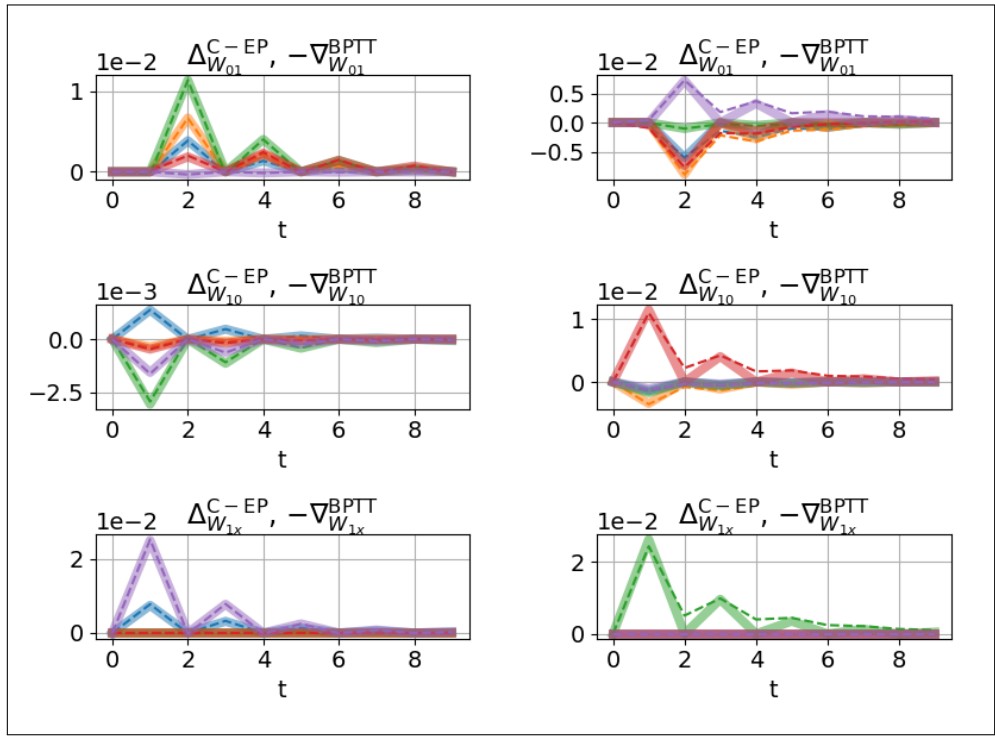

Figure 13: Discrete-Time RNN with asymmetric weights. **Left**: $\Delta_\theta^{\mathrm{C-VF}}(t)$ normalized updates ($\eta = 0$) and $-\nabla_\theta^{\mathrm{BPTT}}(t)$ gradients. **Right**: $\Delta_\theta^{\mathrm{C-VF}}(t)$ normalized updates ($\eta > 0$) and $-\nabla_\theta^{\mathrm{BPTT}}(t)$ gradients.

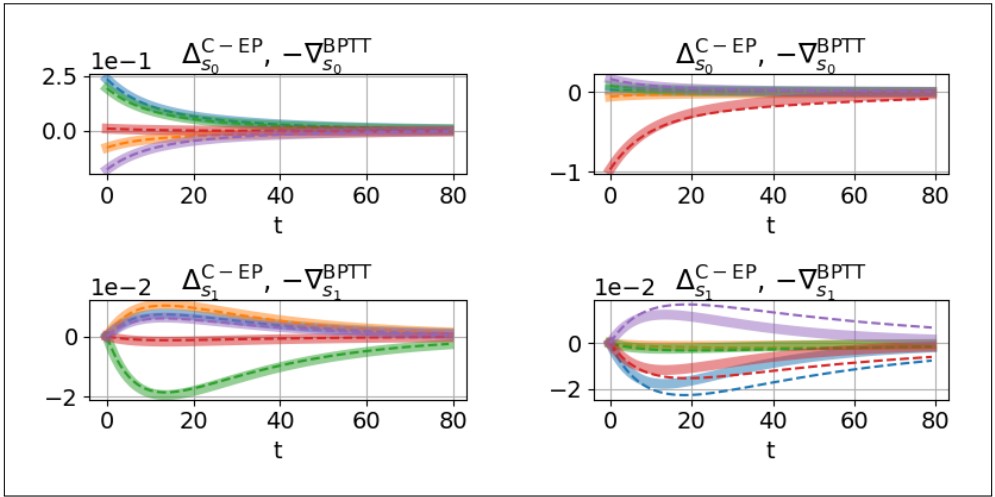

Figure 14: Real-Time RNN with asymmetric weights. **Left**: $\Delta_s^{\mathrm{C-VF}}(t)$ normalized updates ($\eta = 0$) and $-\nabla_s^{\mathrm{BPTT}}(t)$ gradients. **Right**: $\Delta_s^{\mathrm{C-VF}}(t)$ normalized updates ($\eta > 0$) and $-\nabla_s^{\mathrm{BPTT}}(t)$ gradients.

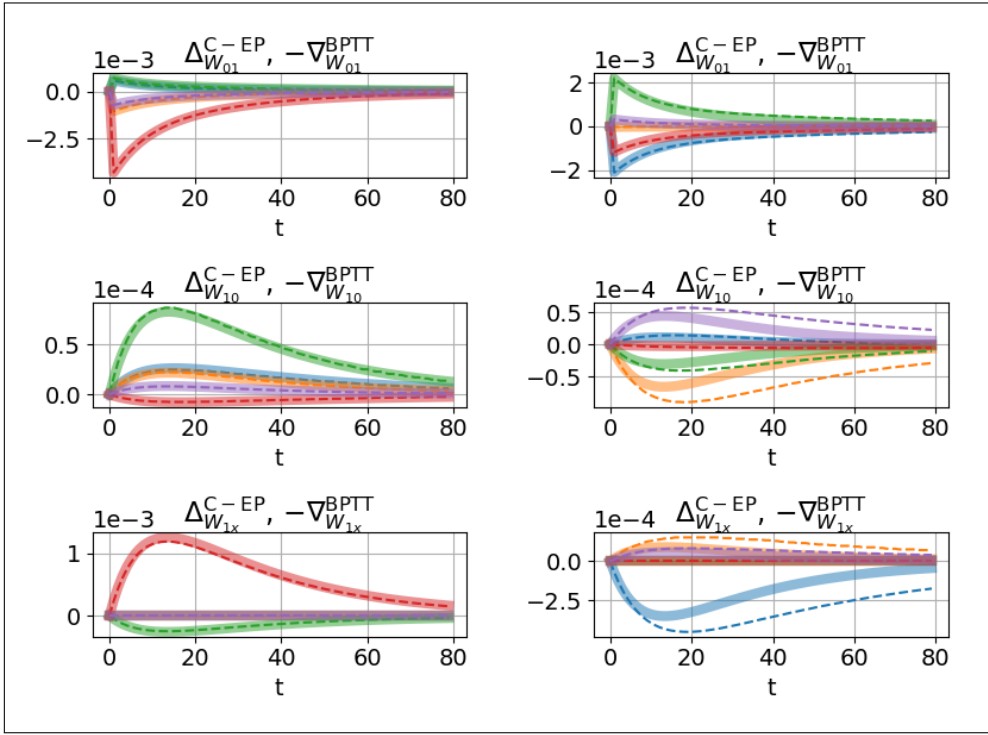

Figure 15: Real-Time RNN with asymmetric weights. **Left**: $\Delta_\theta^{\mathrm{C-VF}}(t)$ normalized updates ($\eta = 0$) and $-\nabla_\theta^{\mathrm{BPTT}}(t)$ gradients. **Right**: $\Delta_\theta^{\mathrm{C-VF}}(t)$ normalized updates ($\eta > 0$) and $-\nabla_\theta^{\mathrm{BPTT}}(t)$ gradients.

# F EXPERIMENTAL DETAILS

## F.1 TRAINING EXPERIMENTS (TABLE 4.1)

**Simulation framework.** Simulations have been carried out in Pytorch. The code has been attached to the supplementary materials upon submitting this work on OpenReview. We have also attached a readme.txt with a specification of all dependencies, packages, descriptions of the python files as well as the commands to reproduce all the results presented in this paper.

**Data set.** Training experiments were carried out on the MNIST data set. Training set and test set include 60000 and 10000 samples respectively.

**Optimization.** Optimization was performed using stochastic gradient descent with mini-batches of size 20. For each simulation, weights were Glorot-initialized. No regularization technique was used and we did not use the persistent trick of caching and reusing converged states for each data sample between epochs as in Scellier and Bengio (2017).

**Activation function.** For training, we used the activation function

$$\sigma(x) = \frac{1}{1 + \exp(-4(x - 1/2))}. \tag{102}$$

Although it is a shifted and rescaled sigmoid function, we shall refer to this activation function as 'sigmoid'.

**Use of a randomized $\beta$.** The option 'Random $\beta$' appearing in the detailed table of results (Table 3) refers to the following procedure. During training, instead of using the same $\beta$ accross mini-batches, we only keep the same *absolute value* of $\beta$ and sample its sign from a Bernoulli distribution of probability $\frac{1}{2}$ at each mini-batch iteration. This procedure was hinted at by Scellier and Bengio (2017) to improve test error, and is used in our context to improve the model convergence for Continual Equilibrium Propagation - appearing as C-EP and C-VF in Table 4.1 - training simulations.

**Tuning the angle between forward and backward weights.** In Table 4.1, we investigate C-VF initialized with different angles between the forward and backward weights - denoted as $\Psi$ in Table 4.1. Denoting them respectively $\theta_f$ and $\theta_b$, the *angle* $\kappa$ between them is defined here as:

$$\kappa(\theta_f, \theta_b) = \cos^{-1}\left(\frac{\mathrm{Tr}(\theta_f \cdot \theta_b^\top)}{\sqrt{\mathrm{Tr}(\theta_f \cdot \theta_f^\top)}\sqrt{\mathrm{Tr}(\theta_b \cdot \theta_b^\top)}}\right),$$

where $\mathrm{Tr}$ denotes the trace, i.e. $\mathrm{Tr}(A) = \sum_i A_{ii}$ for any squared matrix $A$. To tune arbitrarily well enough $\kappa(\theta_f, \theta_b)$, the procedure is the following: starting from $\theta_b = \theta_f$, i.e. $\kappa(\theta_f, \theta_b) = 0$, we can gradually increase the angle between $\theta_f$ and $\theta_b$ by flipping the sign of an arbitrary proportion of components of $\theta_b$. The more components have their sign flipped, the larger is the angle. More formally, we write $\theta_b$ in the form $\theta_b = M(p) \odot \theta_f$ and we define:

$$\Psi(p) = \kappa(\theta_f, M(p) \odot \theta_f), \tag{103}$$

where $M(p)$ is a mask of binary random values {+1, -1} of the same dimension of $\theta_f$: $M(p) = -1$ with probability p and $M(p) = +1$ with probability $1 - p$. Taking the cosine and the expectation of Eq. (103), we obtain:

$$\langle \cos(\Psi(p)) \rangle = p \times -\frac{\mathrm{Tr}(\theta_f \cdot \theta_f^\top)}{\mathrm{Tr}(\theta_f \cdot \theta_f^\top)} + (1 - p) \times \frac{\mathrm{Tr}(\theta_f \cdot \theta_f^\top)}{\mathrm{Tr}(\theta_f \cdot \theta_f^\top)}$$

$$= 1 - 2p$$

Thus, the angle $\Psi$ between $\theta_f$ and $\theta_f \odot M(p)$ can be tuned by the choice of p through:

$$p(\Psi) = \frac{1}{2}(1 - \langle \cos(\Psi) \rangle) \tag{104}$$

**Hyperparameter search for EP.** We distinguish between two kinds of hyperparameters: the recurrent hyperparameters - i.e. $T$, $K$ and $\beta$ - and the learning rates. A first guess of the recurrent hyperparameters $T$ and $\beta$ is found by plotting the $\Delta^{\text{C}-\text{EP}}$ and $\nabla^{\text{BPTT}}$ processes associated to synapses and neurons to see qualitatively whether the theorem is approximately satisfied, and by conjointly computing the proportions of synapses whose $\Delta_W^{\text{C}-\text{EP}}$ processes have the same sign as its $\nabla_W^{\text{BPTT}}$ processes. $K$ can also be found out of the plots as the number of steps which are required for the gradients to converge. Morever, plotting these processes reveal that gradients are vanishing when going away from the output layer, i.e. they lose up to $10^{-1}$ in magnitude when going from a layer to the previous (i.e. upstream) layer. We subsequently initialized the learning rates with increasing values going from the output layer to upstreams layers. The typical range of learning rates is $[10^{-3}, 10^{-1}]$, $[10, 1000]$ for $T$, $[2, 100]$ for $K$ and $[0.01, 1]$ for $\beta$. Hyperparameters where adjusted until having a train error the closest to zero. Finally, in order to obtain minimal recurrent hyperparameters - i.e. smallest $T$ and $K$ possible - we progressively decreased $T$ and $K$ until the train error increases again.

Table 2: Table of hyperparameters used for training. "C" and "VF" respectively denote "continual" and "vector-field", "-#h" stands for the number of hidden layers. The sigmoid activation is defined by Eq. (102).

|  | Activation | T | K | $\beta$ | Random $\beta$ | Epochs | Learning rates |
|---|---|---|---|---|---|---|---|
| EP-1h | sigmoid | 30 | 10 | 0.1 | False | 30 | $0.08 - 0.04$ |
| EP-2h | sigmoid | 100 | 20 | 0.5 | False | 50 | $0.2 - 0.05 - 0.005$ |
| C-EP-1h | sigmoid | 40 | 15 | 0.2 | False | 100 | $0.0056 - 0.0028$ |
| C-EP-1h | sigmoid | 40 | 15 | 0.2 | True | 100 | $0.0056 - 0.0028$ |
| C-EP-2h | sigmoid | 100 | 20 | 0.5 | False | 150 | $0.01 - 0.0018 - 0.00018$ |
| C-VF-1h | sigmoid | 40 | 15 | 0.2 | True | 100 | $0.0076 - 0.0038$ |
| C-VF-2h | sigmoid | 100 | 20 | 0.35 | True | 150 | $0.009 - 0.0016 - 0.00016$ |

Table 3: Training results on MNIST with EP, C-EP and C-VF. "#h" stands for the number of hidden layers. We indicate over five trials the mean and standard deviation for the test error, the mean error in parenthesis for the train error. $T$ (resp. $K$) is the number of iterations in the first (resp. second) phase.

**Full table of results.** Since Table 4.1 does not show C-VF simulation results for all initial weight angles, we provide below the full table of results, including those which were used to plot Fig. 5.

| | Initial $\Psi(\theta_f, \theta_b)$ (°) | Error (%) | | T | K | Random $\beta$ | Epochs |
|---|---|---|---|---|---|---|---|
| | | Test | Train | | | | |
| EP-1h | – | $2.00 \pm 0.13$ | $(0.20)$ | 30 | 10 | No | 30 |
| EP-2h | – | $1.95 \pm 0.10$ | $(0.14)$ | 100 | 20 | No | 50 |
| C-EP-1h | – | $2.85 \pm 0.18$ | $(0.83)$ | 40 | 15 | No | 100 |
| C-EP-1h | – | $2.28 \pm 0.16$ | $(0.41)$ | 40 | 15 | Yes | 100 |
| C-EP-2h | – | $2.44 \pm 0.14$ | $(0.31)$ | 100 | 20 | No | 150 |
| C-VF-1h | 0 | $2.43 \pm 0.08$ | $(0.77)$ | 40 | 15 | Yes | 100 |
| | 22.5 | $2.38 \pm 0.15$ | $(0.74)$ | 40 | 15 | Yes | 100 |
| | 45 | $2.37 \pm 0.06$ | $(0.78)$ | 40 | 15 | Yes | 100 |
| | 67.5 | $2.48 \pm 0.15$ | $(0.81)$ | 40 | 15 | Yes | 100 |
| | 90 | $2.46 \pm 0.18$ | $(0.78)$ | 40 | 15 | Yes | 100 |
| | 112.5 | $4.51 \pm 3.96$ | $(2.92)$ | 40 | 15 | Yes | 100 |
| | 135 | $86.61 \pm 4.27$ | $(88.51)$ | 40 | 15 | Yes | 100 |
| | 157.5 | $91.08 \pm 0.01$ | $(90.98)$ | 40 | 15 | Yes | 100 |
| | 180 | $92.82 \pm 3.47$ | $(92.71)$ | 40 | 15 | Yes | 100 |
| C-VF-2h | 0 | $2.97 \pm 0.19$ | $(1.58)$ | 100 | 20 | Yes | 150 |
| | 22.5 | $3.54 \pm 0.75$ | $(2.70)$ | 100 | 20 | Yes | 150 |
| | 45 | $3.78 \pm 0.78$ | $(2.86)$ | 100 | 20 | Yes | 150 |
| | 67.5 | $4.59 \pm 0.92$ | $(4.68)$ | 100 | 20 | Yes | 150 |
| | 90 | $5.05 \pm 1.17$ | $(4.81)$ | 100 | 20 | Yes | 150 |
| | 112.5 | $20.33 \pm 13.03$ | $(20.30)$ | 100 | 20 | Yes | 150 |
| | 135 | $59.04 \pm 17.97$ | $(60.53)$ | 100 | 20 | Yes | 150 |
| | 157.5 | $77.90 \pm 13.49$ | $(78.04)$ | 100 | 20 | Yes | 150 |
| | 180 | $74.17 \pm 12.76$ | $(74.05)$ | 100 | 20 | Yes | 150 |

## F.2 Why C-EP does not perform as well as standard EP ?

We provide here further ground for the training performance degradation observed on the MNIST task when implementing C-EP compared to standard EP. In practice, when training with C-EP, we have to make a trade-off between:

1. having a learning rate that is small enough so that C-EP normalized updates are subsequently close enough to the gradients of BPTT (Theorem 1),

2. having a learning rate that is large enough to ensure convergence within a reasonable number of epochs.

In other words, the degradation of accuracy observed in the table of Fig. 4.1 is due to using a learning rate that is too large to observe convergence within 100 epochs. To demonstrate this, we implement Alg. 5 which simply consists in using a very small learning rate throughout the second phase (denoted as $\eta_{\text{tiny}}$), and artificially rescaling the resulting weight update by a bigger learning rate (denoted as $\eta$). Applying Alg. 5 to a fully connected layered architecture with one hidden layer, $T = 30$, $K = 10$, $\beta = 0.1$, yields $2.06 \pm 0.13\%$ test error and $0.18 \pm 0.01\%$ train error over 5 trials, where we indicate mean and standard deviation. Similarly, applying Alg. 5 to a fully connected layered architecture with two hidden layers, $T = 100$, $K = 20$, $\beta = 0.5$, yields $1.89 \pm 0.22\%$ test error and $0.02 \pm 0.02\%$ train error. These results are exactly the same as the one provided by standard EP - see Table 3.

---

**Algorithm 5** Debugging procedure of C-EP

*Input*: $x, y, \theta, \beta, \eta, \eta_{\text{tiny}} = 10^{-5}\eta$.
*Output*: $\theta$.

1: $s_0 \leftarrow 0$ ▷ First Phase
2: $\Delta\theta \leftarrow 0$ ▷ Temporary variable accumulating parameter updates
3: **repeat**
4:      $s_{t+1} \leftarrow \frac{\partial \Phi}{\partial s}(x, s_t, \theta)$
5: **until** $s_t = s_*$
6: $s_0^\beta \leftarrow s_*$ ▷ Second Phase
7: **repeat**
8:      $s_{t+1}^\beta \leftarrow \frac{\partial \Phi}{\partial s}\left(x, s_t^\beta, \theta\right) - \beta\frac{\partial \ell}{\partial s}\left(s_t^\beta, y\right)$
9:
10:      $\theta \leftarrow \theta + \frac{\eta_{\text{tiny}}}{\beta}\left(\frac{\partial \Phi}{\partial \theta}\left(s_{t+1}^\beta\right) - \frac{\partial \Phi}{\partial \theta}\left(s_t^\beta\right)\right)$
11:      $\Delta\theta \leftarrow \Delta\theta + \frac{\eta_{\text{tiny}}}{\beta}\left(\frac{\partial \Phi}{\partial \theta}\left(s_{t+1}^\beta\right) - \frac{\partial \Phi}{\partial \theta}\left(s_t^\beta\right)\right)$
12: **until** $s_t^\beta$ and $\theta$ are converged.
13: $\theta \leftarrow \theta - \Delta\theta + \frac{\eta}{\eta_{\text{tiny}}}\Delta\theta$ ▷ Rescale the total parameter update by $\frac{\eta}{\eta_{\text{tiny}}}$

---

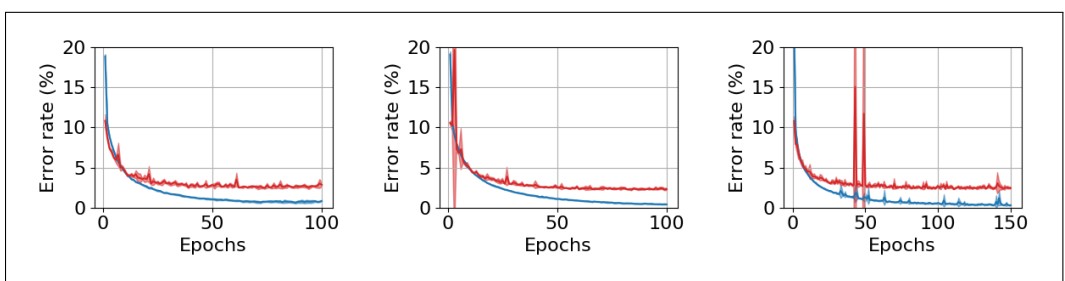

Figure 16: Train and test error achieved on MNIST with Continual Equilibrium Propagation (C-EP) on the Discrete-Time RNN model with symmetric weights. Plain lines indicate mean, shaded zones delimiting mean plus/minus standard deviation over 5 trials. **Left**: C-EP on the fully connected layered architecture with one hidden layer (784-512-10) without beta randomization. **Middle**: C-EP on the fully connected layered architecture with one hidden layer (784-512-10) with beta randomization. **Right**: C-EP on the fully connected layered architecture with two hidden layers (784-512-512-10) without beta randomization.

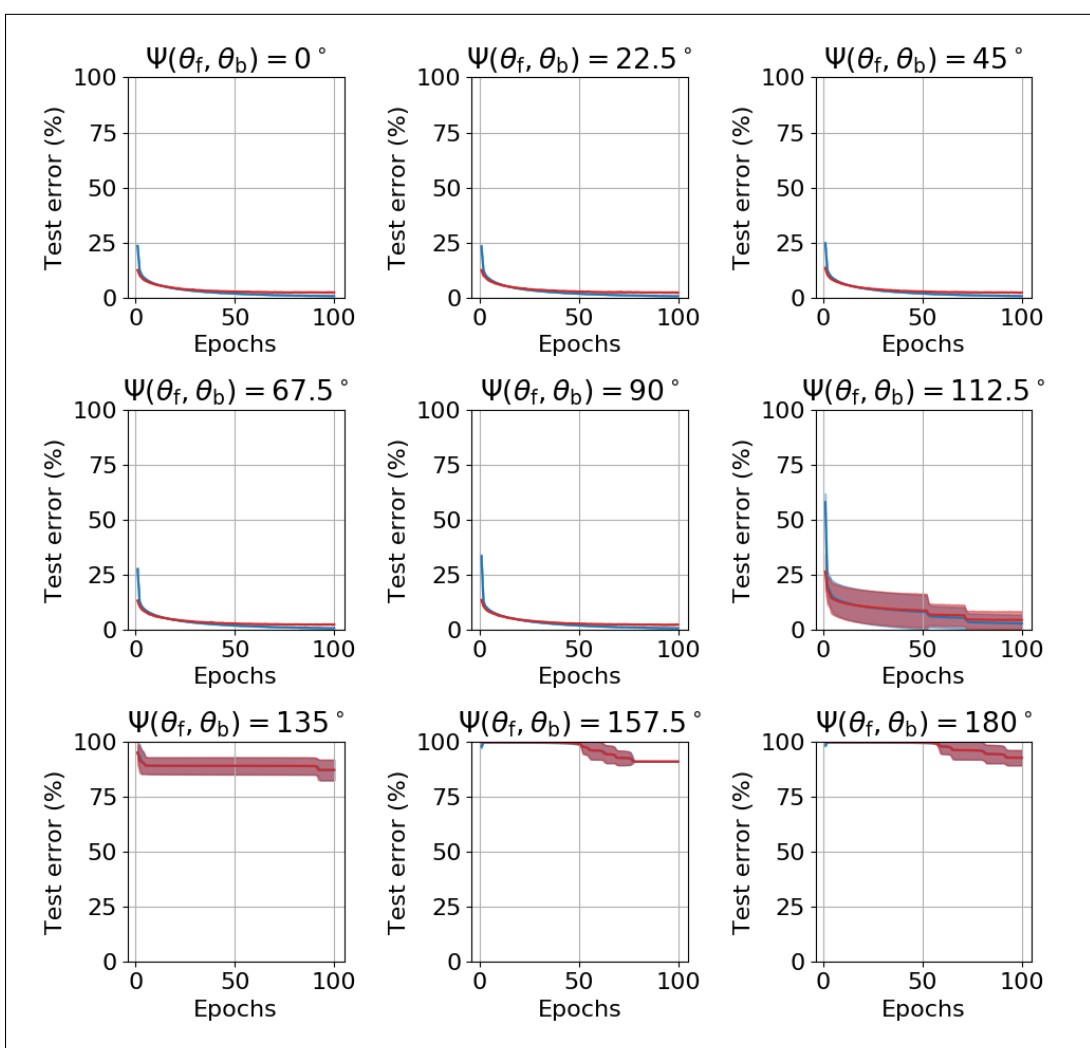

Figure 17: Train and test error achieved on MNIST by Continual Vector Field Equilibrium Propagation (C-VF) on the Discrete-Time RNN model with asymmetric weights with one hidden layer (784-512-10) for different initialization for the angle between forward and backward weights ($\Psi$). Plain lines indicate mean, shaded zones delimiting mean plus/minus standard deviation over 5 trials.

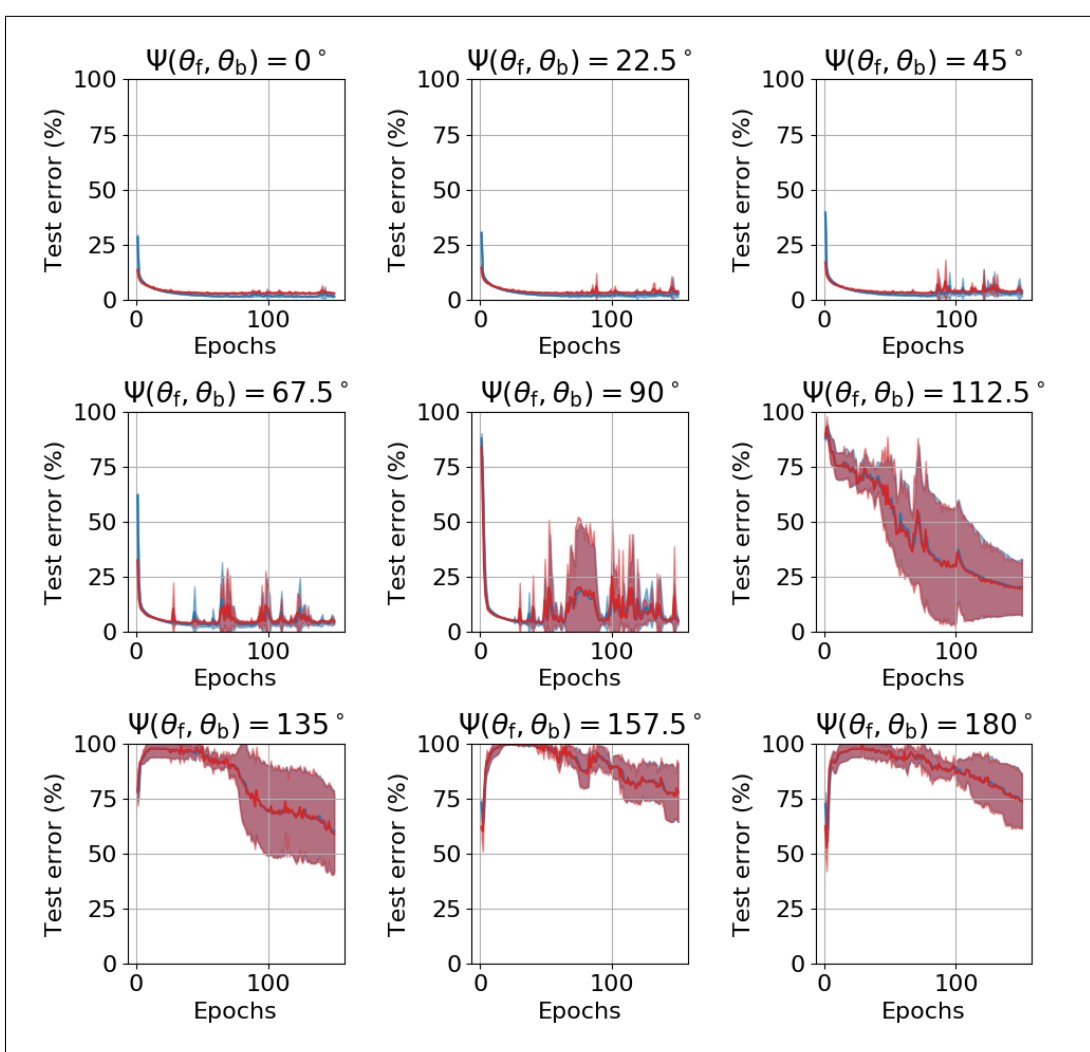

Figure 18: Train and test error achieved on MNIST by Continual Vector Field Equilibrium Propagation (C-VF) on the vanilla RNN model with asymmetric weights with two hidden layers (784-512-512-10) for different initialization for the angle between forward and backward weights ($\Psi$). Plain lines indicate mean, shaded zones delimiting mean plus/minus standard deviation over 5 trials.

