# OpenReview forum: "Equilibrium Propagation with Continual Weight Updates"
_ICLR.cc/2020/Conference — Reject_

### Official Review · AnonReviewer3 · 2019-10-22
**Official Blind Review #3**

**Rating:** 3

**Review:**

The paper is concerned with biologically plausible models of learning. It takes equilibrium propagation -- where updates depend on local spatial information, in the sense that the information is available at each neuron -- and modifies the algorithm so updates are also local in time, thus obtaining C-EP. The key insight is that the updates in EP can be written as a telescoping sum over time points, eq (5).

Making equilibrium propagation more biologically plausible is an interesting technical contribution. But, taking a step back, the setup is misguided. It is true that humans can solve classification problems. And various animals can be trained to do so as well. However, it should be obvious that animals learn to solve these problems by reinforcement learning -- they are literally given rewards like sugar water for correct answers.

MNIST is an unusual dataset with a stark constrast between foreground and background that is far from biologically plausible. I know it has a long and important history in machine learning, but if you are interested in biologically plausible learning then it is simply the wrong dataset to start with from both an evolutionary and developmental perspective. It’s not the kind of problem evolution started with, nor is it the kind of problem human babies start with.

Maybe C-EP can be repurposed into a component of some much larger, biologically plausible learning system that does a mixture of RL and unsupervised learning. Maybe not. The MNIST results provide no indication.

The authors have done a lot of solid work analysing BPTT, RBT, and C-EP. I suspect they are far more interested in understanding and designing efficient mechanisms for temporal credit assignment than they are in biological learning. That work can and should stand on its own feet.


**Experience Assessment:**

I have published one or two papers in this area.

**Review Assessment: Checking Correctness Of Derivations And Theory:**

I assessed the sensibility of the derivations and theory.

**Review Assessment: Checking Correctness Of Experiments:**

I assessed the sensibility of the experiments.

**Review Assessment: Thoroughness In Paper Reading:**

I read the paper thoroughly.

---

> ### Author Response · Authors · 2019-11-09
> **Biological plausibility of C-EP**
>
> We would like to thank the reviewer for his/her comments. Based on this feedback, in the revised version of the paper, we have decided to define explicitly what is meant by biological plausibility in this work, to clarify that continual EP does not aim at being a model of biological learning, and to discuss explicitly the aspects of this algorithm that still  differ from biological learning.
>
> Brains learn using learning rules that have to be local in space and time. Error backpropagation  is particularly non biologically plausible in this regard, as it is fundamentally non local, both in space and time.  Our interest is to propose learning rules that feature the two localities, and this is what we define here as being « biologically plausible ». In this work, we build on EP, which is already local in space, and propose C-EP, which adds locality in time. An important motivation for the development of such minimally biologically-plausible learning rules is that they could be used for the development of extremely energy efficient learning-capable hardware.
>
> We want to be clear that Continual EP does not aim at being a model of biological learning in that it would account for how the brain works or how animals learn. Continual EP does indeed retain considerable differences with actual brain learning. As the Reviewer says, the learning paradigm that is the closest to the way the brain actually learns is Reinforcement Learning. Also, C-EP is evaluated on the MNIST dataset, as the whole current EP literature, which is indeed a conceptual and not a realistic biological task. On the other hand, the use of this task allows a natural bridge with conventional machine learning research. Finally, the equations used in C-EP have no ties with neuroscience experiments.
>
> We propose to make an important overhaul of the introduction and of the discussion of the paper to clarify these points about the nature of our work in the next few days.

---

### Official Review · AnonReviewer1 · 2019-10-24
**Official Blind Review #1**

**Rating:** 3

**Review:**

I think it is an intriguing paper, but unfortunately left me a bit confused. I have to admit is not a topic I'm really versed in, so it might be that this affected my evaluation of the work. But also, as a paper submitted to ICLR I would expect the paper to be self-contained and be able to provide all the details needed.

I do appreciate the authors providing in the appendix the proofs of the other theorems even if they come form other works.


The paper introduces C-EP, an extension of  a previously introduced algorithm EP, such that it becomes biologically plausible. In particular EP is local in space but not in time (you need the steady state of the recurrent state after the first stage at the end of the second stage to get your gradients). I think this is fair, and the need for biological plausibility is well motivated in the beginning of the work.

My first issue is with the proof for the equivalence between EP and C-EP. This is done by taking the limit of eta goes to 0. I think I must be missing something. But the proof relies on eta being small enough such that \theta_i = \theta (i.e. theta does not change). Given this state evolves the same way as for EP, because we basically not changing theta.
Yet the crux of my issue is exactly here. The proof relies on the fact that we don't change theta. So then when you converged on the second phase, isn't theta the same as theta_0? So you haven't actually learned anything!? Basically looking at the delta isn't this just misleading?
Ok lets assume that on the last step you allow yourself to change eta to be non-zero. (I.e. we are just after the delta in theta, and what to show we can get the same delta in theta as EP which is how the proof is phrased). Then in that difference aren't you looking at s_{t+1} and s_t rather than s_{t+1} and s_0, which is what EP would do? In EP you have s^\beta_* - s_*. This is not what you get if you don't update theta and apply C-EP?

I think there might be something I'm missing about the mathematical argument here.

At a higher-level question, we talk about the transition function F as being a gradient vector field, i.e. there exist a phi such that F is d phi/d theta.  Why is this assumption biologically plausable ? Parametrizing gradient vector fields in general is far from trivial, and require very specific structure of the neural implementation of F to be true. Several works have looked at parametrizing gradient vector fields (https://arxiv.org/abs/1906.01563, https://arxiv.org/pdf/1608.05343.pdf) and the answer is that without parametrizing it by actually taking the gradient of a function there is not much of a choice.
Incidentally, here we exploit that  F = sigma (Wx), with W symmetric. This is a paramtrization of a gradient vector field, i.e. of xU, where UU^T =W I think. But if you want to make F deep than it becomes non-trivial to restrict it to gradient vector field. Is the assumption that we never want to move away from vanilla RNNs? And W symmetric is also not biologically plausible. In C-EP you say is not needed to be symmetric, but that implicitly means there exist no phi and everything that follows breaks, no?

I'm also confused by how one has access to d phi / ds and d phi / d theta. Again I feel like I'm missing information and the formalism is not introduced in a way that it is easy to parse. My understand is that you have an RNN that updates the state s. And the transfer function of this RNN is meant to be d phi / ds, which is trues if the recurrent weight is symmetric. Fine. But then why do we have access to d phi/ dtheta? Who is this function? Is the assumption that d s / dtheta is something we can compute in a biologically plausible way? Is this something that is obvious?





**Experience Assessment:**

I do not know much about this area.

**Review Assessment: Checking Correctness Of Derivations And Theory:**

I assessed the sensibility of the derivations and theory.

**Review Assessment: Checking Correctness Of Experiments:**

I assessed the sensibility of the experiments.

**Review Assessment: Thoroughness In Paper Reading:**

I read the paper at least twice and used my best judgement in assessing the paper.

---

> ### Author Response · Authors · 2019-11-09
> **Equivalence between EP and C-EP**
>
> We thank the reviewer for his/her comments.
>
> 1 - Concerning the equivalence between EP and C-EP (Lemma 2, p.~11), which states that $\lim_{\eta \to 0} \Delta_{\theta}^{\rm C-EP}(\eta, \beta, t) = \Delta_{\theta}^{\rm EP}(\beta, t)$.
>
> A) The first point that we want to clarify deals with what we call an `update'.
> In machine learning in general, one usually distinguishes between the error gradient $\frac{\partial L}{\partial \theta}$ and the update $\Delta\theta = \eta \frac{\partial L}{\partial \theta}$, which is the gradient rescaled by a learning rate $\eta$.
> In C-EP in contrast, what we deceivingly call an `update' and denote $\Delta_\theta^{\rm C-EP}(\beta,\eta,t)$ actually corresponds to the gradient, not the update itself.
> To get the actual parameter update in C-EP one needs to rescale $\Delta_\theta^{\rm C-EP}(\beta,\eta,t)$ by $\eta$ ;
> the actual update is $\eta \; \Delta_\theta^{\rm C-EP}(\beta,\eta,t)$, so that $\theta_{t+1}^{\eta,\beta} = \theta_{t}^{\eta,\beta} + \eta \; \Delta_{\theta}^{\rm C-EP}(\eta, \beta, t)$.
> For this reason, when $\eta$ is tiny (or even zero), it is not contradictory that $\theta_{t}^{\eta, \beta}$ does not change while $\Delta_\theta^{\rm C-EP}(\beta,\eta,t)$ is non-zero -- more generally in machine learning it is not incompatible that the update is zero while the gradient is non-zero, if the learning rate is $\eta = 0$.
> In the rest of our answer, to better convey the idea that $\Delta_\theta^{\rm C-EP}(\beta,\eta,t)$ corresponds to a gradient (and not an update in the usual sense),
> we will refer to it as the `forward-time gradient' of C-EP.
> The term `update' will be used to refer to $\eta\Delta_\theta^{\rm C-EP}$.
> We also propose to change the terminology in the whole manuscript (not done yet).
>
> B) Although Lemma~2 holds in the limit $\eta \to 0$,  in practice there is however a trade-off between taking $\eta$ small enough so that the forward-time gradient $\Delta_\theta^{\rm C-EP}(\beta,\eta,t)$ is close enough to $\Delta_\theta^{\rm EP}(\beta, t)$, but not too tiny so that the parameter update $\eta \Delta_\theta^{\rm C-EP}(\beta,\eta,t)$ is not too small to ensure the loss is optimized within a reasonable time (see the bottom of p.6 of the submitted manuscript and Appendix~F.2 p.30). Taking $\eta = 0$ in practice is thus excluded.
> In the proof of Lemma 2 in Appendix A.3, we take $\eta = 0$ because it is mathematically equivalent to taking the limit $\eta \to 0$ (with $\eta >0$), by continuity. We have clarified the proof in the revised version of the manuscript.
>
> C) To understand the equivalence of C-EP and EP, one key thing to have in mind is that if the second phase of EP is run for $K$ steps (i.e. if it takes $K$ steps to get from the first steady state $s_*$ to the second steady state $s_*^\beta$) then the total forward-time gradient (in the sense defined above) of EP is not $\Delta_\theta^{\rm EP}(\beta,K)$ but $\Delta_\theta^{\rm EP}(\beta,0) + \Delta_\theta^{\rm EP}(\beta,1) + \cdots + \Delta_\theta^{\rm EP}(\beta,K)$. To see why this is the case, one has to look at the definition of $\Delta_\theta^{\rm EP}(\beta,t)$ (see Appendix~A.3, Eq.~(18)).
> Now, let us denote $\Delta_\theta^{\rm EP}(\beta,{\rm tot}) = \Delta_\theta^{\rm EP}(\beta,0) +\Delta_\theta^{\rm EP}(\beta,1) + \cdots + \Delta_\theta^{\rm EP}(\beta,K)$ the total gradient of EP, for short.
> If we do the suggested procedure, keeping $\eta = 0$ for the first $K-1$ steps and changing $\eta$ to a positive value at time step $K$, then the effective update of C-EP is $\theta^{\beta,\eta}_{K} - \theta^{\beta,\eta}_{K-1}$ (also equal to $\eta \Delta_\theta^{\rm C-EP}(\beta,\eta,K)$ ). By Lemma 2, this C-EP update is close to $\eta \Delta_\theta^{\rm EP}(\beta,K)$, but not to the total update of EP, which is $\eta \Delta_\theta^{\rm EP}(\beta,{\rm tot})$.
> Conversely, if we keep a constant $\eta$ positive but sufficiently small throughout the second phase, the total parameter update of C-EP at the end of the second phase (after $K$ time steps) is approximately equal to the total parameter update of EP:
>     $\theta^{\beta,\eta}_K - \theta^{\beta,\eta}_0 = \sum_{t=0}^{K-1}(\theta_{t+1}^{\eta, \beta} - \theta_{t}^{\eta, \beta})  = \sum_{t=0}^{K-1} \eta \Delta_{\theta}^{\rm C-EP}(\eta, \beta, t) $
>     $\approx \sum_{t=0}^{K-1} \eta \Delta_{\theta}^{\rm EP}(\beta, t) = \sum_{t=0}^{K-1} \eta \frac{1}{\beta} \left(\frac{\partial \Phi}{\partial \theta}(s_{t+1}^\beta) -\frac{\partial \Phi}{\partial\theta}(s_t)\right) =  \eta\frac{1}{\beta}\left(\frac{\partial \Phi}{\partial \theta}(s_*^\beta) -\frac{\partial \Phi}{\partial\theta}(s_*)\right)$
> To derive the above equation, we have successively used: a telescoping sum, the definition of $\Delta_{\theta}^{\rm C-EP}(\eta, \beta, t)$, Lemma 2, the definition of $\Delta_{\theta}^{\rm EP}(\beta, t)$, and another telescoping sum. We propose to include this equation after the proof of Lemma 2, Appendix A.3, p.11.

---

> > ### Author Response · Authors · 2019-11-09
> > **Properties of the transition function F**
> >
> > 2 - Concerning the properties of the transition function F:
> >
> > A) First of all, in the revised version of the manuscript we are going to clarify the primary goal of our work, which is to address two issues related to the biological plausibility of EP: the first is the fact that the learning rule of EP is not local in time, the second is the requirement of a primitive function  $\Phi$ for the transition function $F$.
> >
> > C-EP solves the first problem but not the second: it still relies on the biologically unrealistic assumption that $F=\frac{\partial \Phi}{\partial s}$. This is precisely this constraint that motivates the second part of our work, where we introduce the C-VF model that gets rid of this assumption.
> >
> > B) In our "C-EP model" with a symmetric weight matrix $W$ (section 4.1), the transition function $F$ (almost) derives from a primitive function $\Phi$.
> > This property is true for any topology (not just a fully connected recurrent network) as long as existing connections have symmetric values: this includes networks with multiple layers (deep networks), in which case the variable $s$ represents the concatenation of all the layers of neurons, and the weight matrix $W$ is a block sparse concatenation of all the layers of weights. More explicitly, denoting the layers of neurons $s^0$, $s^1$, ..., $s^N$, with $W_{n, n+1}$ connecting the layers $s^n$ and $s^{n+1}$ (in both directions), then $s = (s^0, s^1, \dots, s^N)^\top$ and
> >
> > $W =
> > \begin{bmatrix}
> > 0           & W_{01}      & 0           & 0      & 0              & 0         \\
> > W_{01}^\top & 0           & W_{12}      & 0      & 0              & 0         \\
> > 0           & W_{12}^\top & 0           & W_{23} & 0              & 0         \\
> > 0           & 0           & W_{23}^\top & 0      & \ddots         & 0         \\
> > 0           & 0           & 0           & \ddots & 0              & W_{N-1,N} \\
> > 0           & 0           & 0           & 0      & W_{N-1,N}^\top & 0
> > \end{bmatrix}$
> >
> > We propose to add this clarification in Appendix~E.
> >
> > Alternatively, to see why $F$ (almost) derives from a function $\Phi$ in the setting with multiple layers, it is also possible to directly redefine the function $\Phi$ in this specific case: we define  $\Phi = \sum_{n} (s^n)^\top \cdot W_{n, n+1}\cdot s^{n+1}$. In the revised version of the manuscript, we are going to amend Appendix~E, in which the models with multiple layers are detailed, by writing explicitly the form of the function $\Phi$ for each of them.
> >
> > C) In the C-VF model of section 4.1, the weight matrix is no longer assumed to be symmetric, thus there is no primitive function $\Phi$, and therefore Theorem 1 does not apply.
> > Although our study of the C-VF model is mostly experimental, we also prove a generalisation of the GDD theorem that holds in this more general setting (Theorem 2 in Appendix D.2). Fig.5 illustrates this generalisation of the GDD theorem.
> > We have clarified this in the paragraph after Eq.(11) in the revised manuscript.
> >
> > D) We clarify here a point that we had not explained in our manuscript.
> > The theory of our paper (section 3) directly assumes a function $\Phi$ and defines the transition function as $F = \frac{\partial \Phi}{\partial s}$.
> > In the experimental section however (section 4), we proceed the other way around: we first define $F$, then show the existence of a $\Phi$ such that $F \approx \frac{\partial \Phi}{\partial s}$, which $\Phi$ can finally be used to compute the quantities of the form $\frac{\partial \Phi}{\partial \theta}$ required in the learning rule.
> >
> > More concretely, let us consider the case of the C-EP model of section 4.1.
> > We first define the dynamics $s_{t+1} = \sigma(W\cdot s_t)$.
> > This dynamics can be rewritten in the form $s_{t+1} = F(s_t,W)$ with the transition function $F(s,W) = \sigma(W\cdot s)$.
> > In this case, if we define $\Phi(s,W) = \frac{1}{2}s^\top\cdot W\cdot s$, we can compute $\frac{\partial \Phi}{\partial s} = W\cdot s$,
> > and then notice that $F \approx \frac{\partial \Phi}{\partial s}$ if we ignore $\sigma$.
> > Now that we have the analytical expression of $\Phi$, we can also use it to compute $\frac{\partial \Phi}{\partial W}(s,W) = s^\top \cdot s$.
> > Finally we can compute the forward-time gradient of C-EP, which reads $\Delta_W^{\rm C-EP}(\beta,\eta,t) = \frac{1}{\beta} \left( s_{t+1}^{{\beta,\eta}^\top} \cdot s_{t+1}^{\beta,\eta} - s_t^{{\beta,\eta}^\top} \cdot s_t^{\beta,\eta} \right)$.

---

### Official Review · AnonReviewer2 · 2019-10-25
**Official Blind Review #2**

**Rating:** 8

**Review:**

Summary: this paper introduces a new variant of equilibrium propagation algorithm that continually updates the weights making it unnecessary to save steady states. The also mathematically prove the GDD property and show the effectiveness of
 their algorithm (Continual-EP) on MNIST. They also show C-EP is conceptually closer to biological neurons than EP.

This paper tackles an important problem in bridging the gap between artificial neural networks and biological neurons. It is well-motivated and stands well in the literature as it improves its precedent algorithm (EP). The contributions are clear and well-supported by mathematical proofs. The experiments are accurately designed and results are convincing. I recommend accepting this paper as a plausible contribution to both fields.

**Experience Assessment:**

I have read many papers in this area.

**Review Assessment: Checking Correctness Of Derivations And Theory:**

I assessed the sensibility of the derivations and theory.

**Review Assessment: Checking Correctness Of Experiments:**

I assessed the sensibility of the experiments.

**Review Assessment: Thoroughness In Paper Reading:**

I read the paper at least twice and used my best judgement in assessing the paper.

---

> ### Author Response · Authors · 2019-11-09
> **Answer to Review #2**
>
> We thank the reviewer for his/her comments, and are happy that he/she appreciated our work.

---

### Author Response · Authors · 2019-11-15
**Overview of revisions and responses**

We thank the reviewers for their valuable comments, which have help us improve our manuscript - see our revised version.

Based on their feedback, we have now revised our manuscript with the following amendments:

1- To address the request of Reviewer # 3, we have now defined precisely what we meant by "biological plausibility" in our context of study. We have emphasized that the main motivation of our study was not the development of a model of biological learning but to make EP better comply with hardware constraints which in particular require both the locality in space and time of the learning rule. For this purpose, we have amended the abstract, the introduction and the discussion.

2- We have proceeded to a change of terminology in the whole manuscript: the quantity $\Delta_{\theta}^{\rm C-EP}$ is no longer called an 'update' but a 'normalized update' to stress that $\Delta_{\theta}^{\rm C-EP}$ is not the effective parameter update (which is $\eta\Delta_{\theta}^{\rm C-EP}$) but the update normalized by the learning rate $\eta$.
This addresses 1-A) of our answer to Reviewer # 1.

3- We have clarified the proof of Lemma 2 in Appendix A.3, taking the limit $\eta \to 0 $ with $\eta > 0$. This addresses 1-B) of our answer to Reviewer # 1.

4- We have clarified the link between the total parameter update of C-EP and the standard learning rule of EP ($\Delta\theta = \frac{\eta}{\beta}\left(\frac{\partial\Phi}{\partial s}(s^\beta_*) - \frac{\partial\Phi}{\partial s}(s_*) \right)$) after the derivation of Lemma 2 in Appendix A.3. This addresses 1-C) of our answer to Reviewer # 1.

5- We have clarified in the introduction that our work addresses two issues of EP: first that its learning rule is not local in time, second that it relies on the requirement of a primitive function $\Phi$ for the transition function $F$; the first issue is addressed with the C-EP algorithm, the second with C-VF algorithm. This addresses 2-A) of our answer to Reviewer # 1.

6- In section 4, we now make a clear distinction between the training algorithms (C-EP and C-VF) and the models they train. What was previously called the 'C-EP model' has become the 'Vanilla RNN with symmetric weights trained with C-EP'. Likewise, the 'C-VF model' has become the 'Vanilla RNN with asymmetric weights trained with C-VF'. We have also stressed after Eq.(12) (of the revised manuscript), after introducing the Vanilla RNN model with asymmetric weights trained with C-VF that although in this case the dynamics do not derive from a primitive function $\Phi$, Theorem 1 can be generalized to this setting, by referring to the related Appendix where this generalization is derived. This addresses 2-C) of our answer to Reviewer # 1.

7- We have explained in details why our vanilla RNN model with symmetric weights trained with C-EP described in Section 4.1 extends to deep architectures with any number of layers with symmetric connections. Also, we have explicitly written the primitive function $\Phi$ for all our models. For this purpose we have added mathematical details in Appendix E describing all the models used in the papers, on a simple example and we now refer to this Appendix in Section 4.1. This addresses 2-B) and 2-D) of our answer to Reviewer # 1.

---

### Decision · Program_Chairs · 2019-12-19

**Decision:**

Reject

**Comment:**

Main content: paper introduces a new variant of equilibrium propagation algorithm that continually updates the weights making it unnecessary to save steady states. T

Summary of discussion:
reviewer 1: likes the idea but points out many issues with the proofs.
reviewer 2: he really likes the novelty of paper, but review is not detailed, particularly discussing pros/cons.
reviewer 3: likes the ideas but has questions on proofs, and also questions why MNIST is used as the evaluation tasks.
Recommendation: interesting idea but writing/proofs could be clarified better. Vote reject.